# DropBlot: single-cell western blotting of chemically fixed cancer cells

Yang Liu [1,2] ✉ & Amy E. Herr [1,3] ✉

Archived patient-derived tissue specimens play a central role in understanding disease and developing therapies. To address specificity and sensitivity shortcomings of existing single-cell resolution proteoform analysis tools, we introduce a hybrid microfluidic platform (DropBlot) designed for proteoform analyses in chemically fixed single cells. DropBlot serially integrates droplet-based encapsulation and lysis of single fixed cells, with on-chip microwell-based antigen retrieval, with single-cell western blotting of target antigens. A water-in-oil droplet formulation withstands the harsh chemical (SDS, 6 M urea) and thermal conditions (98 °C, 1-2 hr) required for effective antigen retrieval, and supports analysis of retrieved protein targets by single-cell electrophoresis. We demonstrate protein-target retrieval from unfixed, paraformaldehyde-fixed (PFA), and methanol-fixed cells. Key protein targets (HER2, GAPDH, EpCAM, Vimentin) retrieved from PFA-fixed cells were resolved and immunoreactive. Relevant to biorepositories, DropBlot profiled targets retrieved from human-derived breast tumor specimens archived for six years, offering a workflow for single-cell protein-biomarker analysis of sparing biospecimens.

An estimated 1 billion archived tumor tissues are housed in biorepositories and medical centers[1]. Archived tissues make retrospective studies possible and retrospective studies are needed to understand diseases and develop therapies. For example, retrospective studies of tissues powered the development of trastuzumab (Herceptin®), arguably one of the most effective targeted cancer therapies ever developed[2]. To preserve cellular morphology and prevent the degradation of proteins during storage, archived tissues are chemically fixed (i.e., formalin[3]; paraformaldehyde (PFA)[4]; methanol[5]; formalin-fixed, paraffin-embedded (FFPE)[6]). Cell fixation is a crucial component of clinical pathology and biomedical research, allowing stabilization of proteins during extended archiving of the cellular material. Prior to analysis of archived cells and tissue, these fixed cells require harsh pretreatments to partially restore antigen immunoreactivity for subsequent measurement (i.e., antigen retrieval)[7–9] typically by immunoassay. Although much remains to be learned about the design of high-quality tissue-fixation protocols and the mechanisms by which antigen immunoreactivity is restored, fixed

and archived biospecimens have substantially influenced clinical medicine and will continue to do[10,11].

For analysis of fixed tissues and even single fixed cells, immunohistochemistry (IHC)[12] and immunocytochemistry (ICC)[13] are widely used in pathology and biomedical research labs. Fluorescence and colorimetric stains report protein-target presence, localization, and distribution (e.g., human epidermal growth factor receptor 2 (HER2)[14], programmed death ligand 1 (PD-L1)[15]). For analysis of suspensions of fixed cells, flow cytometry is a potent tool for rapid protein analysis and cell sorting (i.e., size, shape, and biomolecular profile using fluorescently labeled antibody probes)[16,17]. Flow cytometry can process more than 10,000 cells per second with multiplexity of up to ~20 protein targets[16]. While useful, flow cytometry requires a large starting number of cells (>10,000 cells), and has limited detection specificity due to inadequate probes, probe and spectral signal overlap, and cellular autofluorescence[16]. In contrast, mass spectrometry (MS) is a powerful protein detection tool that does not require antibody probes, but MS has limitations and is not appropriate for all protein-

[1]Department of Bioengineering, University of California, Berkeley, CA 94720, USA. [2]School of Chemical, Materials and Biomedical Engineering, University of Georgia, Athens, GA 30602, USA. [3]Chan Zuckerberg Biohub, San Francisco, CA 94158, USA. ✉e-mail: liuy@uga.edu; aeh@berkeley.edu

based questions[18]. Top-down proteomics struggles with single-cell detection[18], and bottom-up MS obscures proteoform stoichiometry by requiring protein digests as input[19]. Antibody-based MS methods offer high-resolution analyses but inherently depend on specific antibody probes[20], similar to immunoassays. Proteoform Imaging Mass Spectrometry (PiMS) provides increased spatial resolution but cannot analyze proteins >70 kDa, limiting the use of PiMS to certain cancer-related protein targets[21].

Prior to any analytical stage, sample preparation is a critical stage with microfluidic techniques making inroads into the preparation of fixed cells. Droplet-based[22] and reaction chamber-based[23] microfluidic systems offer an enclosed compartment to facilitate cell incubation, lysis, and antigen extraction under even harsh conditions, in preparation for analysis of diverse cell types and myriad fixation conditions. Microfluidic large-scale integration (mLSI) platforms[24], single-cell barcode chips (SCBCs)[25], and droplet-based cell screening & sorting[26] reduce the number of cells required for protein analysis, as compared to flow cytometry and conventional MS. New measurement methods[27] may improve protein detection sensitivity and specificity in fixed cell and tissue samples. When considering analysis of biospecimens archived in biorepositories, limiting consumption of sparingly available sample masses—while maximizing detection sensitivity and specificity—can emerge as a central design tradeoff.

While not optimized for analysis of sparingly available biospecimens, slab-gel immunoblotting is a workhorse targeted-proteomic method with a specificity that is sufficient to detect protein proteoforms and protein complexes. A type of immunoblot, western blotting, couples protein polyacrylamide gel electrophoresis (PAGE) with subsequent immunoassays[28]. Slab-gel immunoblotting of fixed samples has been reported[29,30], often pooling tissue or cell samples to enhance detection sensitivity. Our lab introduced a suite of immunoblotting tools optimized for individual unfixed (fresh) cells[31,32], including for analysis of human-derived dissociated solid tumors[33] and circulating tumor cell specimens[34]. For fresh or fresh-frozen clinical specimens, our research group has introduced single-cell immunoblotting[31,32]. In addition to obtaining cellular- and subcellular-resolution protein profiles, precision immunoblotting allows researchers to more directly and quantitatively compare cell-to-cell levels of protein expression. Since the expression of each protein target is measured for each microwell-isolated cell, expression can be readily normalized on a per-cell or even per-cell-volume basis. To foster technology translation and support concurrent analyses of 100's–1000's of cells, we design 'open-microfluidic' chips that do not incorporate enclosed microchannels or pneumatic control (pumps, valves). Instead, the open-fluidic chips are similar to a mini-gel layered on a standard microscope slide, but with arrays of open microwells stippled into the open-faced gel. The microwells are sized to isolate single dissociated tumor cells and circulating tumor cells for subsequent imaging, lysis, and ultra-rapid protein PAGE through the polyacrylamide gel surrounding each microwell. While relevant to readily lysed fresh cells, the open-microwell design supports only brief cell-lysis durations (<1 min, <50 °C) before the lysate dilutes and diffuses out of the microwell, thus making the readily translatable tools irrelevant to the preparation of fixed cells that require long-duration (>60 min) and harsh antigen retrieval conditions[35,36]. Benchmarking of contemporary single-cell proteomic analysis techniques and associated performance tradeoffs are summarized in Supplementary Table 1.

Consequently, we introduce a hybrid microfluidic tool—called DropBlot—that extends the relevance of open-fluidic chip design to the preparation and analysis of single fixed cancer cells, and aims to conserve the use of precious, archived biorepository specimens. DropBlot is designed to provide target specificity suitable for proteoforms and protein complexes with single-cell resolution. These protein-target classes often lack specific antibody probes. As a corollary design goal, the DropBlot assay is designed to limit consumption of sparingly available

archived, fixed biospecimens, hence the use of microfluidic design versus slab-gel assay formats. While not the subject of this study, the DropBlot reported here aims to create a foundation for extension of the DropBlot to analysis of rare cells, such as our previously reported single-cell immunoblotting tools for analysis of patient-derived circulating tumor cells (CTCs)[34] and individual cells (blastomeres) derived from single two-cell and four-cell murine embryos[37–39]. DropBlot combines droplet-based, single-cell sample preparation with open-fluidic, single-cell western blotting for 100's–1000's of chemically fixed cells, as is directly relevant to archived biospecimens. To restore antigen immunogenicity prior to electrophoresis and immunoassays, we report the design of stable water-in-oil (W/O) droplets to encapsulate single fixed cells and support cell lysis at high temperatures (100 °C) and long incubation periods (1–2 hr). Lysate-containing droplets are sedimented into open microwells. We employ electrotransfer to inject solubilized protein targets from each microwell-encapsulated droplet into the PAGE chip, for protein separation, in-gel protein blotting (covalent immobilization of protein to gel polymer), and subsequent immunoassays. We develop the DropBlot workflow using two breast cancer cell lines and a suite of chemical fixation conditions. Each protein target is first individually identified by immunoreactivity from fresh-cell (unfixed) lysate and then under each fixation condition for a well-characterized pair of cancer cell lines. We expect each protein target to vary in electromigration behavior depending on the fresh or fixed state of the originating cell, thus methodical assay development and target-identity tracking are utilized. After validating the identity and immunoreactivity of each protein target, we next apply DropBlot to investigate a pilot cancer-protein panel composed of an epithelial marker (EpCAM), a mesenchymal marker (vimentin, VIM), and human epidermal growth receptor 2 (HER2) in a pilot group of breast cancer patient-derived cell samples. Our results demonstrate western blotting of single PFA-fixed cells, and form the basis of a modular sample-preparation and analysis tool, adaptable to the panoply of cell-fixation conditions used in biorepositories.

## Results and discussion
### Overview of DropBlot
To perform single-cell western blotting on antigen targets retrieved from chemically fixed cells, we sought to seamlessly integrate two disparate functions into one assay workflow: (1) antigen retrieval from single fixed cells using droplet microfluidic technology with (2) single-cell PAGE and immunoblotting of the retrieved antigen targets using a planar array of single-cell PAGE separations on an open microfluidic device (Fig. 1a). The hybrid DropBlot design is comprised of three typically independent microfluidic modes (i.e., cell-laden droplets, microwells, and planar chip separations) as a way to unify the diverse chemical, thermal, electrical, and mechanical conditions required at each stage of the preparation and analysis process into one integrated workflow (Fig. 1b–f).

### Design of droplets for stability to the chemical, thermal, and mechanical requirements of DropBlot
We sought to achieve four performance goals by selecting a water-in-oil (W/O) chemistry for the droplets and a microfluidic H-junction for droplet generation. First, we sought to co-encapsulate single cells and a surfactant-containing antigen-retrieval buffer, in such a way that antigen solubilization would occur after the droplet is fully formed. The performance goal seeks to ensure sufficient antigen retrieval while minimizing antigen leakage from the droplet. Second, we sought droplet chemistry that maintains stability under harsh antigen-retrieval conditions, which include the presence of surfactants (SDS), exposure to elevated temperature (>95 °C), and incubation for longer than 1 hr (see Methods)[40,41]. Third, we sought a stable droplet formulation robust to mechanical handling and seating of said droplets into the open-fluidic array of microwells. To achieve a high fraction of droplets that contain a single cell, we adopted a technique for deterministic droplet encapsulation of inertially ordered cells[42]. Fourth, we

 

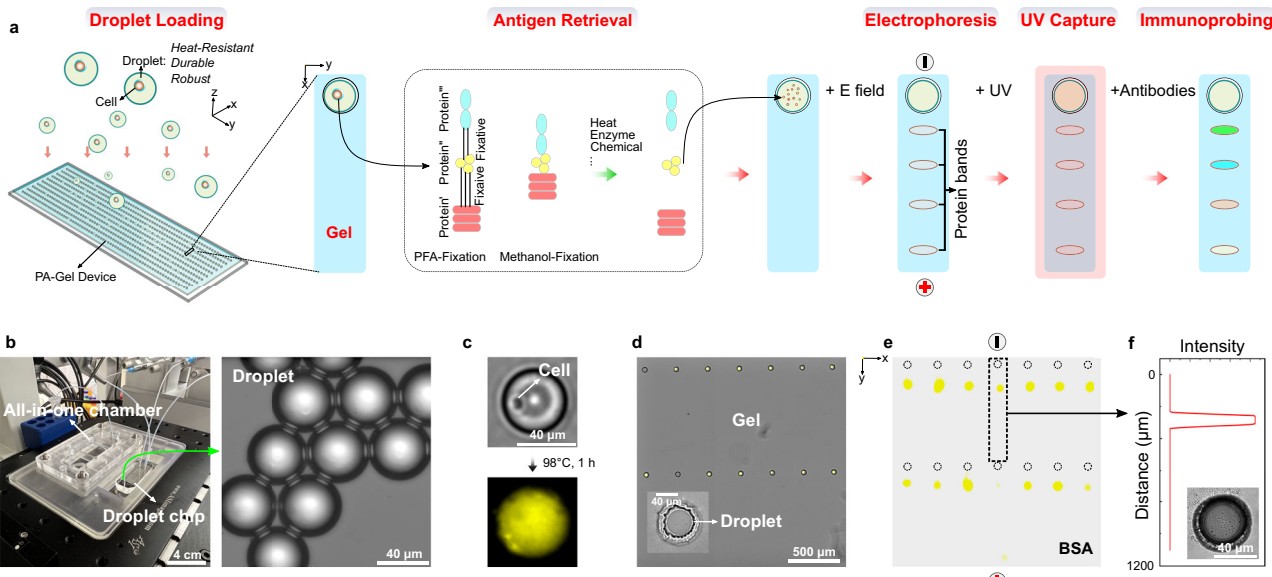

**Fig. 1 | Design, operation, and characterization of DropBlot, a hybrid microfluidic platform that couples single fixed-cell sample preparation in droplets to single-cell protein immunoblotting in a planar, open-fluidic chip.**
**a** Conceptual schematic of the droplet-based single-cell preparation stage interfacing with the device-based single-cell western blotting stage. The tandem assay is designed to solubilize immunoreactive antigen targets from individual chemically fixed cells for subsequent single-cell immunoblotting. **b** Photo of the hybrid-assay assembly, which mates the droplet-generation chip (left) with a PAGE chamber (left). Brightfield image of droplets for cell preparation stage (right). **c** Fluorescence micrographs of GFP-expressing MCF7 lysing in a W/O droplet. False-color yellow is GFP fluorescence signal. **d** Brightfield micrograph microwell array and abutting PAGE regions for single-cell western blotting. False-color yellow is AF488-labeled BSA signal. Inset: micrograph of a droplet-containing microwell. **e** Fluorescence micrograph of an array of PAGE endpoint analyses of AF488-labeled BSA protein standard ($\Delta t_{PAGE} = 20$ s, $E = 40$ V/cm). Anode and cathode orientation is as marked. **f** Background-subtracted fluorescence intensity (AFU) of one PAGE lane. Inset: brightfield micrograph of a droplet-containing microwell after PAGE. **b**–**f** Data are representative of three independent experiments with similar results. Source data are provided as a Source Data file.

sought a stable-droplet formulation amenable to electrotransfer of solubilized antigen out of each microwell-encapsulated droplet and into a PAGE gel proximal to each microwell.

Given these four target-performance specifications, we opted for a water-in-oil (W/O) droplet chemistry, with droplets generated at an H-junction to allow the co-introduction of single cells with the antigen-retrieval buffer. Our first step was systematically optimizing the droplet generation function for single cell per droplet occupancy, by considering channel geometry, flow rates of the continuous and dispersed phases, and the initial concentration of the cell suspension (MCF7 cells, Supplementary Fig. 1). To ensure a high fraction of droplets contained a single cell, we designed the droplet-loading chip to order cells using a serpentine geometry that achieves inertial focusing of the cells for subsequent ordered loading of droplets (Supplementary Fig. 2), as has been reported recently[42–46]. Single-cell occupancy was achieved in $79.5 \pm 6.1\%$ ($n = 3$) of droplets generated under the following empirically determined conditions: droplet diameter = $\varnothing_{droplet} = 40$–$60$ μm; volumetric flow rate of dispersed phase = $Q_{dispersed} = 0.2$–$20$ μL/min; volumetric flow rate of continuous phase = $Q_{continuous} = 0.2$–$100$ μL/min; starting concentration of cell suspension = $1.0$–$6.0 \times 10^6$ cells/mL; loading time: <15 min.

After establishing baseline single cell per droplet occupancy with the H-junction, we sought to establish stable droplets compatible with our selected antigen-retrieval protocols. For the droplet design rules, we considered two major factors: (i) properties of the surfactants used in the continuous phase (mineral oil)[47,48] and (ii) properties of the dispersed phase (dual lysis and antigen-retrieval buffer which includes SDS, cell suspension)[49,50]. In the continuous phase, surfactants can reduce interfacial tension and prevent droplets from coalescing[47]. For example, mineral oil supplemented with 0.5–5.0% (v/v) of Span® 80 nonionic surfactant results in stable W/O emulsions[51,52]. The dispersed phase—especially when containing SDS or other surfactants—also influences droplet stability[53,54]. As is shown in Fig. 2a, a dispersed phase

containing increasing SDS concentration decreases droplet stability (i.e., $Q_{dispersed} = 10$ μL/min, $Q_{continuous} = 15$ μL/min) as expected, due to destabilization of hydrophilic surfactants in the dispersed phase[50]. To determine a suitable formulation for our DropBlot performance goals, we screened a panel of formulations for both the continuous (Span® 80 concentrations = 0.5–5% (v/v)) and dispersed phases (SDS concentrations = 0.1–2% (w/v)). We scrutinized the bulk stability of the droplet formulations over a 3-hr incubation at elevated temperature (80–100 °C) by employing brightfield imaging of a homogeneous suspension of droplets in a tube (Fig. 2b, c). Any visually detectable phase separation of the suspension (with the concomitant visible formation of immiscible layers) indicates droplet breakage. We observed no observable phase separation of materials layers for the formulation consisting of a continuous phase containing 2% (v/v) Span® 80 and a dispersed phase containing 0.5% SDS (w/v) ($\varnothing_{droplet} = 50$ μm). Further, we observed no notable change in the number of droplets by droplet enumeration microscopy ($\Delta t = 0$ hr, $n_{droplets} = 657 \pm 31$; $\Delta t = 3$ hr, $n_{droplets} = 588 \pm 28$; $n = 3$). Given these findings, we next sought to understand the cell-lysis efficacy for model cell lines. Model cell lines are important for iterative assay development, because primary cells can be precious and highly variable. When engineered to express fluorescent reporter proteins, model cell lines also allow imaging-based assay optimization. Using fluorescence microscopy inspection, we observed cell lysis of GFP-expressing MCF7 cells within 5 min with a 0.5% (w/v) SDS formulation in the dispersed phase (Fig. 2d, e; $95.1 \pm 1.2\%$ lysis efficiency; $n_{droplets} = 1000$; $n = 3$). Important to our mechanical robustness performance goal, brightfield visual inspection showed lysate-filled droplets remained largely intact after deposition into the microwells ($92.3 \pm 4.7\%$ yield of intact droplets; $n_{microwells} = 560$; $n = 3$).

To ensure droplet stability under the chemical conditions required by the antigen-retrieval function, we next evaluated protein leakage from the selected W/O droplet formulation using a continuous

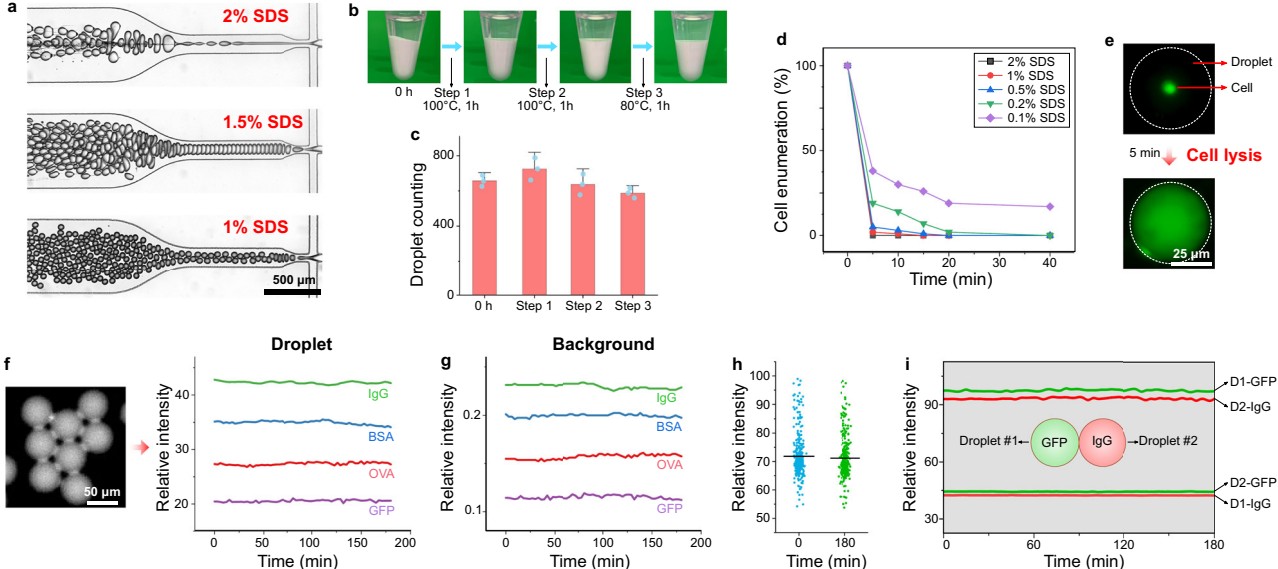

**Fig. 2 | Design and generation of stable droplets as required for fixed-cell lysis and antigen retrieval, followed by droplet loading into individual microwells for electrotransfer to lysate PAGE. a** Brightfield micrographs show droplet generation with 0.5–2.0% (w/v) sodium dodecyl sulfate (SDS) in the dispersed medium ($V_{dispersed}$ : $V_{continuous}$ = 10:15 μL/min). Continuous phase [Span® 80] = 2% (v/v). **b** Brightfield image time series assessing droplet stability for the incubation sequence. Top layer (transparent): mineral oil; Bottom layer (opaque): droplets. (Droplet diameter $\varnothing_{droplet}$ = 50 μm, 0.5% (w/v) SDS) **c** Droplet enumeration during the incubation sequence in **b**. Droplets were incubated and imaged on a glass slide with a hydrophobic surface. The error bars represent the standard deviation of the mean (n = 3 measurements). **d** Fraction of MCF7 cells lysed as a function of SDS concentration at 95 °C ($\varnothing_{droplet}$ = 50 μm) **e** Fluorescence micrograph showing 'before' and 'after' lysis of a GFP-expressing MCF7 cell encapsulated in a W/O droplet (RT, 0.5% SDS (w/v)). Data are representative of three independent experiments with similar results. **f** Fluorescence micrograph of a droplet loaded with Alexa-Fluor 555 labeled BSA. (right): Mean fluorescence intensity monitoring of ~300 individual droplets, each loaded with AF488-IgG, AF555-BSA, AF647-OVA, and GFP. **g** Mean fluorescence intensity monitoring of background signal for the droplets in **f**. **h** Mean fluorescence intensity monitoring of 300 single droplets loaded with AF555-BSA. The lines represent the mean value (n = 300 droplets). **i** Mean fluorescence intensity of paired droplets loaded with GFP (left droplet, green shading) and AF555-IgG (right droplet, red shading). Flux of fluorescence signal is considered into and out of both droplets. GFP intensity in Droplet #1 (D1-GFP); IgG intensity in Droplet #2 (D2-IgG); GFP intensity in Droplet #2 (D2-GFP); IgG intensity in Droplet #1 (D1-IgG). Source data are provided as a Source Data file.

phase of 2% (v/v) Span® 80 and a dispersed phase containing 0.5% (w/v) SDS. A fluorescently labeled protein standard solubilized in antigen-retrieval buffer (SDS: 0.5% (w/v)) was loaded into the droplets, with fluorescence microscopy used to monitor the total integrated fluorescence intensity of individual droplets over time. We observed no notable decrease in fluorescence intensity during the 180-min monitoring period at room temperature (Fig. 2f). The background intensity remained nearly constant over 3 hr (Fig. 2g), and the intensity distribution of individual droplets did not notably differ before and after the experiment (Fig. 2h). Additionally, we observed no notable fluorescence crosstalk between neighboring droplets loaded with protein ladder targets labeled with spectrally distinct fluorophores (Fig. 2i). These findings suggest a formulation for stable droplets suitable for the preservation of target proteins with negligible target loss or crosstalk over the course of a 180-min experiment. Taken together, we adopted the described conditions as a starting point for developing the DropBlot for fixed cancer cells.

## Design of droplets for stability to the electrical and mechanical requirements of DropBlot

We next sought to determine suitable conditions for (i) mechanical deposition of the cell-lysate-containing droplets onto the microwell array and (ii) electrotransfer of soluble protein targets from microwell-encapsulated droplets into the proximal polyacrylamide gel for single-cell western blotting of retrieved antigens. To understand the impact of physical handling of droplets, W/O droplets containing lysate from single unfixed cells were mechanically deposited onto the open-fluidic chip used for the DropBlot protein analysis stage. Droplets were sedimented onto the surface of the open gel, and gentle washing removed droplets that did not sediment into microwells

(Supplementary Fig. 4). Inspection of the microwells by fluorescence microscopy reported ~95.1 ± 3.1% (n = 3 chips) of microwells occupied with a single droplet after 15 min of settling. We observed that intact droplets loaded into the microwells did not noticeably coalesce with the polyacrylamide gel walls of the microwells.

We next aimed to understand both the feasibility and efficiency of electrotransfer as a low-dispersion means to introduce solubilized (retrieved) antigens from the microwell-encapsulated droplet into the microwell-abutting protein PAGE analysis gel. While the electrotransfer concept is based on an approach our group has used for electrophoretic analysis of single-cell lysates[31], DropBlot presents a notable difference in the presence of an immiscible phase between the starting location of the antigen targets, that being in a droplet, and the endpoint location, that being in a proximal molecular sieving hydrogel.

To understand this more chemically and geometrically complex sample-injection configuration, we asked how the presence of an immiscible phase and the placement of each droplet within a microwell affects the injection of protein target into the sieving matrix, with particular interest in injection efficiency and dispersion (Fig. 3a, b).

Using a 2D numerical simulation that we created in COMSOL®, we considered the immiscible and miscible phases (i.e., oil layer thickness) and geometry (i.e., electric field strength, microwell size, relative position between droplet and microwell). The conductivity of the oil layer was assumed to be similar to mineral oil, σ = 0.175 S/m (note: the conductivity of Span® 80 is negligible compared to mineral oil[55,56]). Simulations suggest that a voltage drop across the droplet results in an applied electric field within the core of the droplet and through the immiscible oil layer (Fig. 3c). Establishing a continuous−if not uniform−electric field across all phases of the configuration is the minimum necessary attribute needed to use electrotransfer to move a charged

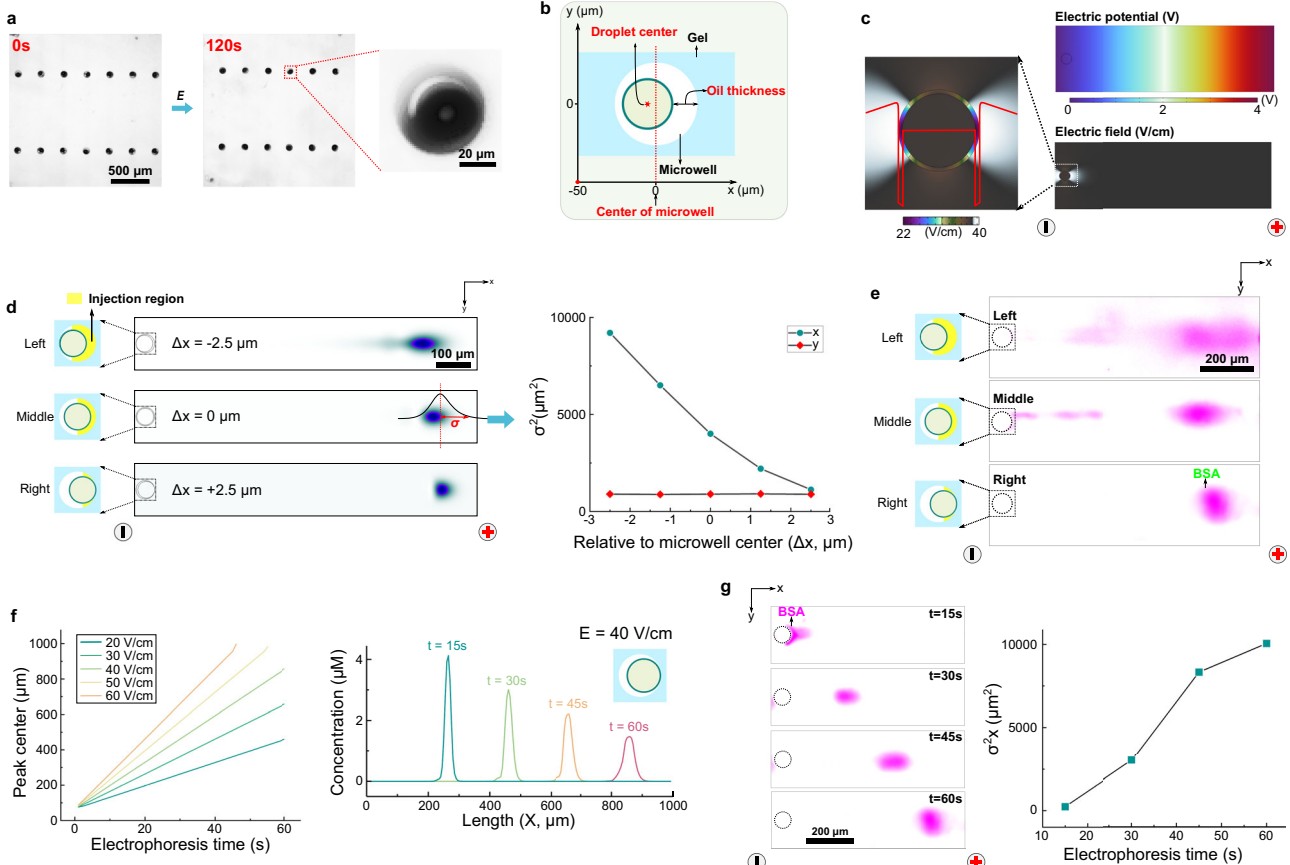

**Fig. 3 | Simulation and validation of protein-target electrotransfer from a microwell-encapsulated W/O droplet into abutting PAGE gel. a** Droplet stability test under electric field (*E*, field strength: 40 V/cm). The droplets remain intact after 120 s electrophoresis. **b** Schematic showing a top view (*x-y*) of the planar DropBlot model for study of electrotransfer of protein lysate from droplet, through oil region, into thin-layer polyacrylamide gel for PAGE. Microwell center is located at *x* = 0 μm. Droplet *y* position is aligned with *y*-position of the microwell. **c** Simulation showing the applied potential and resultant electric field around geometry shown in **b**. The red line represents electric field strength along the *x*-direction of the PAGE separation gel abutting each microwell. **d** Simulation of the migration distance (left) and concentration profiles (right) of protein ladder component BSA as a function of various droplet positions relative to microwell center (Left: Δx = −2.5 μm; Middle: Δx = 0 μm; Right: Δx = +2.5 μm). Yellow shading demarcates injection region; σ = BSA peak width; ∅$_{microwell}$ = 50 μm; ∅$_{droplet}$ = 45 μm; *E* = 40 V/cm; Δt$_{PAGE}$ = 60 s. **e** Companion fluorescence micrographs of protein ladder component AF555-BSA for the same conditions presented in simulations reported in **d**. **f** Simulation of Right-aligned droplet (Δx = +2.5 μm) configuration and BSA PAGE electromigration as a function of elapsed PAGE time (*E* = 40 V/cm) and applied electric field strength. **g** Companion fluorescence micrographs of AF555-BSA electromigration along the PAGE separation axis for a Right-aligned droplet (Δx = +2.5 μm) configuration. Resultant empirical relationship between concentration profile and electrophoresis time. Source data are provided as a Source Data file.

analyte from within a droplet, through the oil layer, and into an abutting polyacrylamide gel.

The 2D simulations further informed us that the applied electric field distribution across the droplet-to-microwell interface would create an analyte-stacking interface (at the O/W interface), and an analyte-destacking interface (as the species electromigrate through the oil layer). A second analyte-staking interface would be expected upon analyte electromigration into the polyacrylamide gel interface forming the walls of each microwell. To understand and control these discrete regions, we define an 'injection region' as the span between the right edge of a microwell-encapsulated droplet and the right edge of the encapsulating microwell. When the anode is located to the right along the separation axis, the protein-migration direction is defined as +x (i.e., negatively charged protein electromigrates from left to right, Fig. 3d). For a fixed microwell diameter, the thickness of the injection region is determined by the droplet diameter and the relative position of the droplet in the microwell (Δx). The retention time in the injection region, $t_{RT}$, is estimated by $t_{RT} = L_{inj}/(E_{inj} \times \mu_{inj})$ where $L_{inj}$ is the length of the injection region (cm), $E_{inj}$ is the strength of the applied electric field (V/cm) in the injection region, and the $\mu_{inj}$ is the electrophoretic mobility (cm²/(V·s)) of the charged analyte

through the injection region (e.g., $\mu_{BSA}$= 125 μm²/(V·s); Supplementary Fig. 5)

Given the nonuniform electric field distribution across the droplet-to-microwell geometry—and the anticipated concomitant impact on peak dispersion through stacking and destacking—we arrive at design rules including: (i) the oil layer should be as thin as possible to mitigate destacking, while keeping in mind droplet stability and (ii) the droplet should be seated proximal to the injection region at the head of the gel sieving matrix, again to minimize destacking. For example, reducing the droplet size or the thickness of the oil layer, and/or moving the droplet from Δx = 2.5 μm to Δx = −2.5 μm, will induce longer retention time and peak broadening during PAGE (Fig. 3d, Supplementary Figs. 6–9).

To experimentally validate the design rules and optimize the performance of the electroinjection and subsequent PAGE separation, we experimentally scrutinized 45-μm diameter BSA-containing droplets seated in 50-μm diameter microwells. Tilting the assembly allowed us to position each droplet adjacent to the gel lip on the rightmost side of the microwell, thus minimizing the injection region length. Once the droplets were positioned in the microwells, an electric field was applied (*E* = 40 V/cm) and BSA was observed

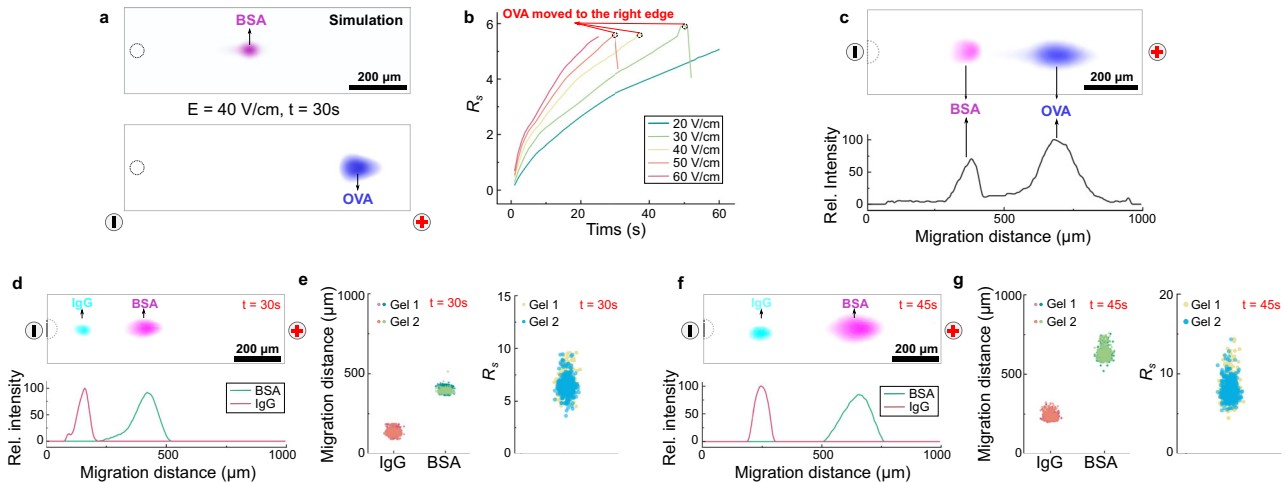

**Fig. 4 | DropBlot assay development utilizing purified protein standards.**
**a** Simulation of PAGE electromigration for BSA and OVA ($\Delta t_{PAGE}$ = 30 s; $E$ = 40 V/cm). **b** Simulation of separation resolution ($R_s$) of BSA and OVA as a function of applied $E$. Open circles indicate when OVA peak reaches terminus of PAGE separation lane. **c** Fluorescence micrograph of PAGE electromigration for AF555-BSA (pink) and AF647-OVA (blue) ($\Delta t_{PAGE}$ = 30 s; $E$ = 40 V/cm) above fluorescence intensity profile. **d** Fluorescence micrograph of PAGE electromigration for AF555-BSA (pink) and AF488-IgG (blue) ($\Delta t_{PAGE}$ = 30 s; $E$ = 40 V/cm) above fluorescence intensity profile. **e** Comparison of migration distance (left) and separation

resolution (right) for PAGE of AF555-BSA and AF488-IgG standard proteins (n = 500 PAGE lanes; N = 2 devices; $\Delta t_{PAGE}$ = 30 s; $E$ = 40 V/cm). **f** Fluorescence micrograph of PAGE electromigration for AF555-BSA (pink) and AF488-IgG (blue) ($\Delta t_{PAGE}$ = 45 s; $E$ = 40 V/cm) above fluorescence intensity profile. **g** Comparison of migration distance (left) and separation resolution ($R_s$, right) for PAGE of AF555-BSA and AF488-IgG standard proteins (n = 500 PAGE lanes; N = 2 devices; $\Delta t_{PAGE}$ = 45 s; $E$ = 40 V/cm). For all results: $\varnothing_{microwell}$ = 50 µm; $\varnothing_{droplet}$ = 45 µm; Right-aligned droplet ($\Delta x$ = +2.5 µm) configuration. Source data are provided as a Source Data file.

electromigrating out of the droplet, through the minimal injection region, and then into the PAGE sieving matrix. Across a range of droplet positions relative to the microwell center and lip, we observed impacts on electromigration and injection dispersion (Fig. 3e). The observed behaviors were dependent on the droplet oil-layer thickness in the injection region. Once in the sieving matrix, the migration and dispersion of the BSA peak were proportional to the elapsed PAGE time and applied electric field strength (Fig. 3f, g). At the completion of PAGE and photocapture of each protein peak to the gel matrix, we observed by brightfield microscopy that the originating droplets—now devoid of fluorescently labeled BSA tracer—remained intact in each microwell, thus suggesting the robust operation of the selected droplet formulation and overall handling approach under the mechanical and electrical requirements of DropBlot.

We next sought to assess PAGE protein separation performance after electrotransfer of protein sample from the microwell-encapsulated W/O droplets using soluble, fluorescently labeled protein standards (OVA, 43 kDa, $D_{inj}$ = 3.23 µm²/s, $D_{gel}$ = 4.65 µm²/s; BSA, 66 kDa, $D_{inj}$ = 2.40 µm²/s, $D_{gel}$ = 3.45 µm²/s; Fig. 4a–c). By both simulation and experiment, we observed full resolution of OVA from BSA in 30 s (8%T, 3.3%C gel; $E$ = 40 V/cm). The separation resolution ($R_s$ = [$\Delta x$/ (0.5 × (4 $s_1$ + 4 $s_2$))]) where $\Delta x$ is peak-to-peak displacement and 4s is the width of neighboring peaks 1 and 2, respectively) was proportional to the electric field strength and elapsed separation time, as expected, with a slightly higher $R_s$ predicted by simulation ($R_s$ = 4.9) than observed by experiment ($R_s$ = 3.8). Experiments report electrophoretic mobilities of $\mu_{BSA}$= 3590 µm²/(V·s) and $\mu_{OVA}$= 5830 µm²/(V·s) (Supplementary Fig. 5). Electroinjection of a large protein species (fluorescently labeled IgG, 150 kDa) was also feasible, as was resolution of the BSA and IgG (Fig. 4d–g).

### DropBlot analysis of fresh (not fixed) cancer cells
We next sought to scrutinize the preparation and analysis of unfixed cells, testing the sample preparation and integration functions of DropBlot. To assess the relationship between antigen-retrieval buffer formulation (including SDS) and the immunoreactivity of any retrieved antigen, we scrutinized fresh cells from well-studied cultured cell lines

and the protein targets epithelial cellular adhesion molecule (EpCAM, 35–40 kDa) and intermediate filament protein (VIM, 57 kDa). We anticipate the chemical state of each originating cell will impact the degree of antigen retrieval for each protein target. Antigen retrieval means, most obviously, the amount of antigen material recovered from a chemically fixed cell. Importantly, and more subtly, antigen retrieval also depends on the degree of recovery of immunoreactivity, which in turn depends on the physicochemical properties of each retrieved antigen species. In this latter aspect, we anticipate that fixation-induced alterations in protein physicochemical properties will inherently affect electromigration. Consequently, we start by establishing protein-target identity in the simplest to the most complex matrix conditions using a modified 'spiked recovery' immunoassay development process to account—as much as possible—for matrix effects anticipated in human-derived and chemically fixed cell specimens[57]. Here, that means we start with target antigen measurement from a clear buffer, progressing next to antigen measurement from fresh cell lines, then retrieval from fixed cells from the same cell lines, and finally move to consider retrieval of a panel of protein targets in fixed patient-derived tumor specimens. We make the assumptions that SDS binds proteins with a constant mass ratio (i.e., 1.4 to 1 (SDS: protein)[58] or 3 to 1 (SDS: protein)[59], each 45-µm diameter droplet contains ~240 fg of SDS, and that each mammalian cell contains ~100 fg of protein. We reach a working conclusion that the selected droplet volume ($\varnothing$ = 45 µm, volume = 47.8 pL) and the selected antigen-retrieval buffer (0.5% (w/v) SDS) provide a mass of SDS sufficient to coat all protein molecules from each single cell. As shown in Fig. 5, solubilization, electrotransfer, and PAGE analysis were demonstrated using this combination. Immunoreactivity was sufficiently recovered for each protein target, based on successful endpoint immunoblotting.

Expanding the repertoire of cell preparation conditions accessible with DropBlot, we explored the possibility of including 6 M urea in the 0.5% (w/v) SDS antigen-retrieval buffer (Fig. 5a, b, Supplementary Fig. 10). Urea, a strong chaotropic agent, can break hydrogen bonds and unfold hydrophobic protein regions by disrupting hydrophobic interactions[60]. SDS-based antigen-retrieval buffer supplemented with a high concentration of urea (e.g., 6 M) can solubilize a variety of

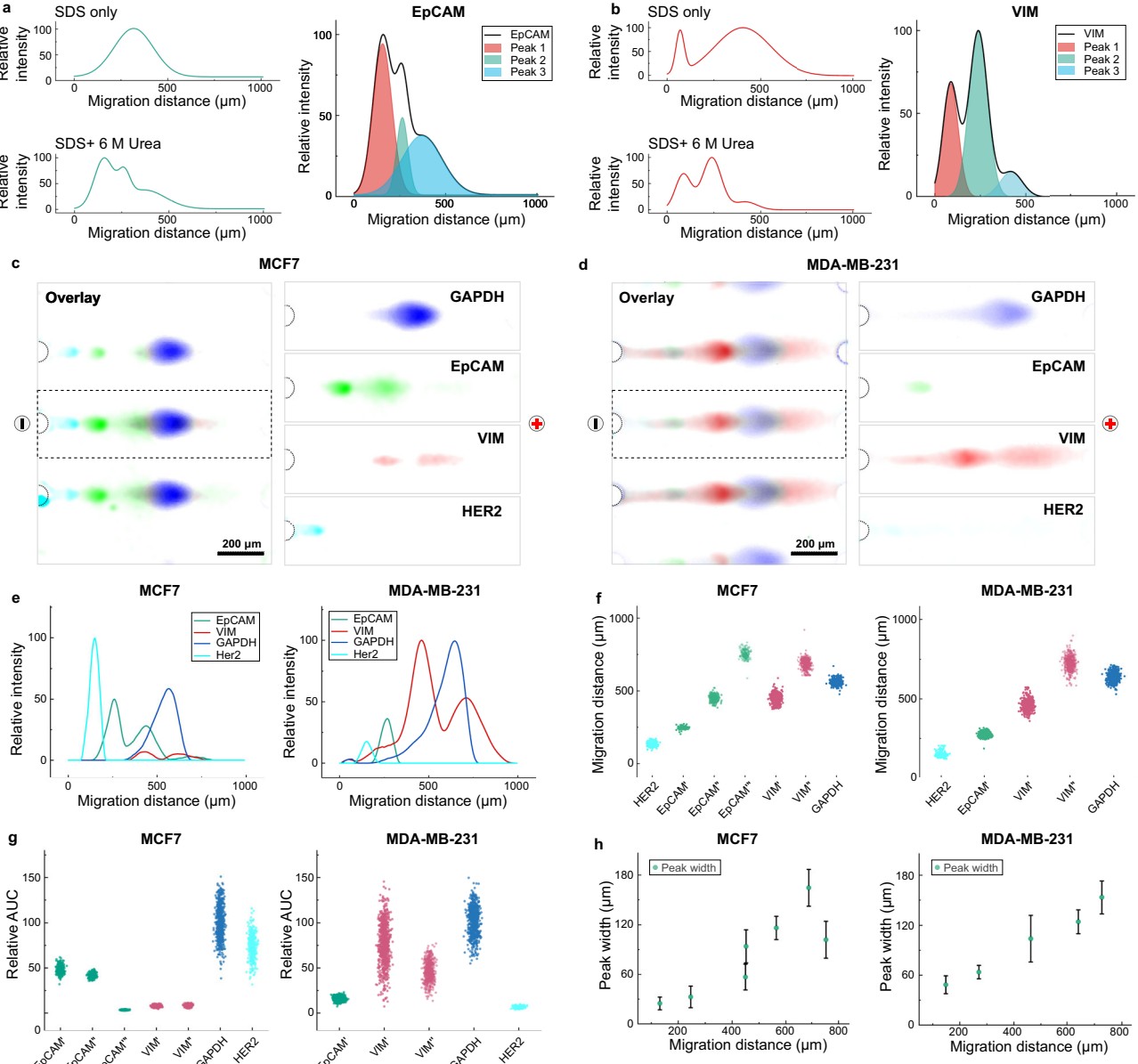

**Fig. 5 | DropBlot assay development utilizing single cells from two unfixed breast cancer cell lines, MCF7 and MDA-MB-231. a** Fluorescence intensity profile for representative EpCAM PAGE separations from two unfixed MCF7 cells lysed with 0.5% SDS antigen-retrieval buffer without (top) and with (bottom) a 6 M urea supplement ($\Delta t_{PAGE}$ = 30 s; $E$ = 40 V/cm). Fluorescence intensity plot (right) shows Gaussian fitting assuming three overlapping EpCAM peaks in SDS + 6 M urea antigen-retrieval buffer formulation. **b** Fluorescence intensity profile for representative VIM PAGE separations from two unfixed MDA-MB-231 cells lysed with 0.5% SDS antigen-retrieval buffer without (top) and with (bottom) a 6 M urea supplement ($\Delta t_{PAGE}$ = 30 s; $E$ = 40 V/cm). Fluorescence intensity plot (right) shows Gaussian fitting assuming three overlapping VIM peaks in SDS + 6 M urea antigen-

retrieval buffer formulation. Fluorescence micrographs for single-cell PAGE of **c** unfixed MCF7 and **d** unfixed MDA-MB-231 cells for the protein targets EpCAM (green), mesenchymal marker VIM (red), human epidermal growth factor receptor 2 (HER2, cyan), and glycolytic enzyme GAPDH (blue) ($\Delta t_{PAGE}$ = 30 s; $E$ = 60 V/cm). **e** Fluorescence intensity profiles for unfixed cells analyzed in **c** and **d**. **f** Migration distance analysis of respective protein targets from single-cell PAGE analysis of MCF7 (left) and MDA-MB-231 cells (right) (n = 1000 PAGE lanes; $\Delta t_{PAGE}$ = 30 s; $E$ = 60 V/cm). **g** Fluorescence area under curve (AUC) for PAGE analyses from **f**, n = 1000 PAGE lanes. **h** PAGE migration distance and peak width for PAGE analyses from **f**; The error bars represent standard deviation of the mean (n = 1000 PAGE lanes). Source data are provided as a Source Data file.

proteoforms by reducing detergent micelles and breaking detergent-protein complexes[61,62]. We observed that an antigen-retrieval buffer (0.5% (w/v) SDS) supplemented with 6 M urea can resolve additional EpCAM and VIM proteoforms during PAGE of single unfixed cells, with no obvious detrimental effect of the urea supplement on droplet stability (Supplementary Fig. 11).

After establishing the DropBlot system as suitable for the analysis of protein targets from unfixed single cells, we analyzed our multiplexed cancer-protein panel by immunoblotting the targets EpCAM, VIM, endogenous protein GAPDH, and human epidermal growth factor

receptor 2 (HER2). In both a human breast epithelial line (MCF7) and a triple-negative breast cancer line (MDA-MB-231), DropBlot successfully completed unfixed cell and protein sample preparation, then supported immunoblotting of the four targets as shown in Fig. 5c, d. We observed no cross-reactivity between the antibody probes utilized and the four targets from the cancer-protein panel.

Consistent with previous research, we observed that the epithelial cell line, MCF7, had a high expression of EpCAM and HER2 and low expression of VIM, relative to a mesenchymal cell line (MDA-MB-231), which had a high expression of VIM and low expression of EpCAM and

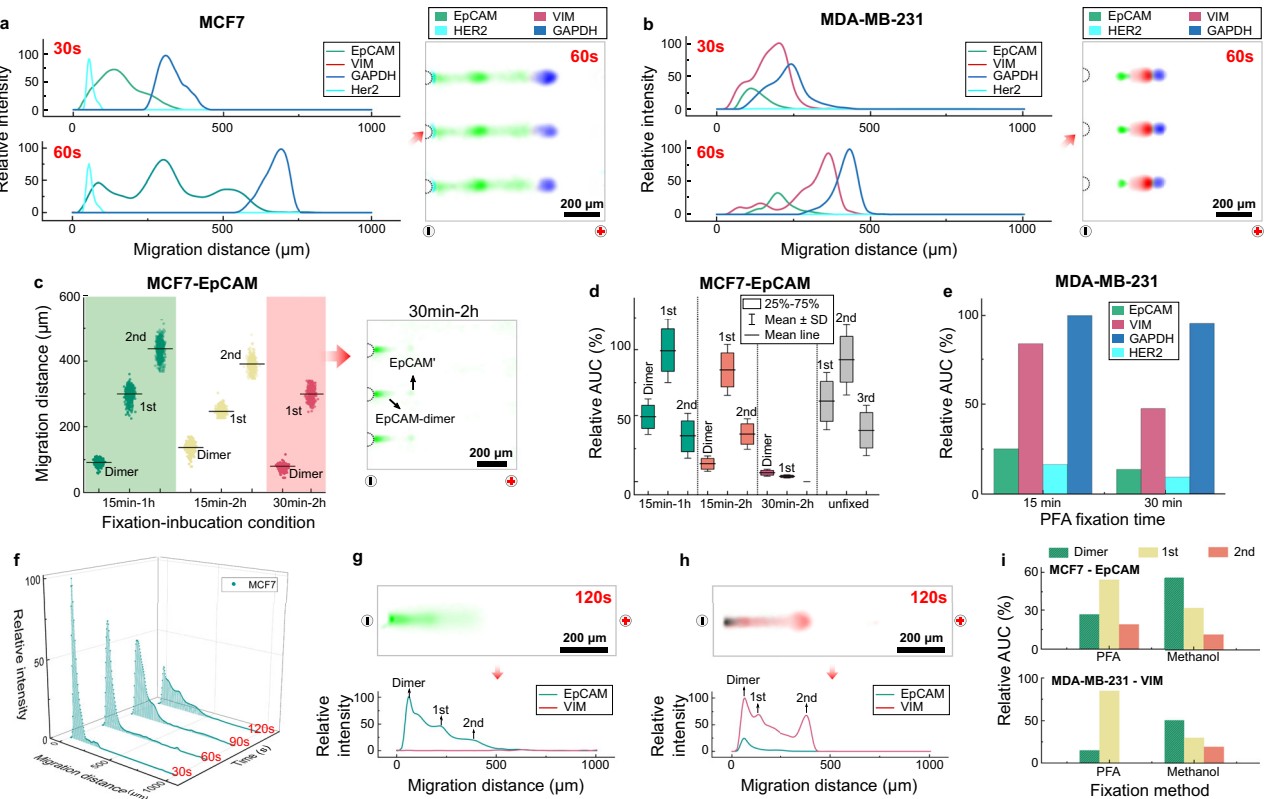

**Fig. 6 | DropBlot assay development utilizing single PFA- and methanol-fixed cancer cells. a** Fluorescence intensity plots and companion fluorescence micrographs of DropBlot analysis of PFA-fixed MCF7 cells. **b** Same as **a** for PFA-fixed MDA-MB-231 cells. **c** Single-cell western blot EpCAM migration distance for three fixation-incubation conditions ($\Delta t_{fixation}$ = 15 or 30 min; $\Delta t_{incubation}$ = 1.0 or 2.0 h at 98 °C) with representative fluorescence micrographs from three PFA-fixed MCF7 cells treated with $\Delta t_{fixation}$ = 30 min; $\Delta t_{incubation}$ = 2.0 h at 98 °C. The lines represent the mean value (n = 1000 PAGE lanes). **d** Fluorescence area under curve (AUC) of single-cell western blot protein peaks for PFA-fixed MCF7 cells ($\Delta t_{fixation}$ = 15 or 30 min). On each box plot, the central mark indicates the mean value, the bottom and top edges of the box indicate the interquartile range (IQR), and the whiskers represent the standard deviation (n = 1000 PAGE lanes). **e** Same as **d** but for PFA-fixed MDA-MB-231 cells ($\Delta t_{fixation}$ = 15 or 30 min). **f** Fluorescence intensity from single-cell western blots of EpCAM from methanol-fixed MCF7 cells for $\Delta t_{PAGE}$ = 30, 60, 90, and 120 s. **g** Fluorescence micrograph from single-cell western blot of EpCAM and VIM from methanol-fixed MCF7 cell above fluorescence intensity profile. **h** Same as **g** for a methanol-fixed MDA-MB-231 cell. **i** Fluorescence area under curve (AUC) of single-cell western blot protein peaks for both cell types and fixation conditions. PFA conditions: $\Delta t_{fixation}$ = 15 min; $\Delta t_{incubation}$ = 1.0 h at 98 °C; $\Delta t_{PAGE}$ = 60 s. Methanol conditions: $\Delta t_{fixation}$ = 15 min; $\Delta t_{incubation}$ = 1.0 h at 98 °C; $\Delta t_{PAGE}$ = 120 s. For all results, PAGE E = 60 V/cm. Source data are provided as a Source Data file.

HER2. The epithelial-to-mesenchymal transition (EMT) can alter the expression of EpCAM, VIM, and HER2[63,64]. We further observed three resolved EpCAM peaks in the lysate of single MCF7 cells, while MDA-MB-231 cells exhibited one detectable peak (Fig. 5e–h). We attribute the cell-line-dependent EpCAM expression to different proteoforms of EpCAM (Supplementary Table 2). In the intricate interplay of cellular distinctions, DropBlot emerges as a discerning tool, capable of delineating between cell types based on their unique fingerprint proteins and proteoforms (Supplementary Fig. 12). Notably, the H1299 lung cancer cell line is demarcated from PBMCs by the distinct presence of CD45 and VIM[65], while discriminating between MDA-MB-231 cells and stromal fibroblasts can rely on nuanced variations in VIM and fibroblast activation protein (FAP) expression levels and proteoforms. Although FAP is also expressed on MDA-MB-231, the expression level and types of proteoforms vary[66].

### DropBlot analysis of fixed cancer cells

We scrutinized the DropBlot technology for retrieval and analysis of cancer-related proteins from two types of fixed cancer cells: paraformaldehyde (PFA) and methanol (see Methods). We scrutinized PFA and methanol as fixation chemistries, because the two chemistries offer different chemical fixation mechanisms (Supplementary Table 3). PFA induces covalent bonds between molecules, effectively binding the molecules together to form an insoluble network that changes the mechanical characteristics of the cell surface[67]. In contrast, methanol is understood to denature and precipitate proteins without the formation of covalent bonds[68].

With these different fixation options in mind, we started by encapsulating fixed cells from each of the well-studied cell lines in the W/O droplets. Each cell was incubated in a droplet with antigen-retrieval buffer at 98 °C for 60–120 min. After being subjected to the full DropBlot workflow, immunoblotting of the single fixed-cell lysates showed that protein targets are detectable from PFA-fixed cancer cell lines (MCF7 and MDA-MB-231, fixation time: 15 min). During PAGE, we further observed protein electromigration as being ~50% slower for targets retrieved from PFA-fixed cells as compared to the respective fresh cancer cell lines (Fig. 6a, b). Reduced electromigration of PFA-fixed cells versus fresh cells is expected because the chemical-fixation chemistry is known to form an insoluble network of molecules within cells by creating new covalent molecular bonds[67]. Based on this PFA-fixation mechanism, PFA-fixed protein species would reasonably have a larger Stokes radius and concomitantly lower electrophoretic mobility (Supplementary Fig. 13), as compared to unfixed (denatured) protein targets. For DropBlot analysis of PFA-fixed cells, we explored the impact of increasing the fixation time (15 vs. 30 min) and increasing the antigen-retrieval incubation time (120 min) and observed that increasing the fixation time (30 min) dramatically reduced the number and intensity of detectable protein peaks (Fig. 6c–e, Supplementary

Fig. 14). Extended PFA fixation can increase the crosslinking strength between proteins and lipids, making antigen retrieval difficult[69]. Further, we sought to achieve a balance between enhanced antigen-target retrieval with elevated incubation temperatures versus deleterious effects that are exacerbated at elevated temperatures in the presence of urea (i.e., carbamylation, protein aggregation)[70,71]. To mitigate carbamylation during antigen retrieval in the presence of 6 M urea, we reduced incubation temperatures (below 37 °C) and acidified the urea solution by adding 100 mM HCl. Acidification helps impede the formation of isocyanic acid, thereby minimizing carbamylation. However, we were unable to arrive at acceptable antigen retrieval performance for DropBlot protocols that included 6 M urea, thus deferring optimization of urea conditions to future assay development efforts.

Similarly, we scrutinized methanol-fixed MCF7 cells and observed that, while EpCAM was detectable by single-cell immunoblotting, electromigration was even slower than the PFA-fixed cells (Fig. 6f). We further observed elevated protein signals near the microwells (Fig. 6g, h, Supplementary Fig. 15), suggesting the presence of large protein dimer molecules or even protein aggregates. Recall, we anticipate that the physicochemical properties of each retrieved antigen (including electromigration, degree of immunoreactivity, and amount of protein material solubilized) will differ depending on the antigen and the chemical conditions of the originating cell, thus the use of a methodical assay development process that starts with fresh-cell lysate from well-characterized cell lines, then assays a suite of chemical fixation conditions for those same cell lines and protein targets, before moving to target identification in chemically fixed, human-derived biospecimens[57].

We further compared the area under the curve (AUC) of EpCAM and VIM in both MCF7 and MDA-MB-231 cells, using both PFA- and methanol-fixation chemistries. Figure 6i illustrates the distribution of EpCAM and VIM proteoforms. In PFA-fixed MCF7 cells, the protein expression profile consisted of: ~26.7% EpCAM-dimer, 54.3% EpCAM', and 19.0% EpCAM" proteoforms. The protein expression profile of methanol-fixed MCF7 cells consisted of: ~56.0% EpCAM-dimer, 32.2% EpCAM', and 11.8% EpCAM" proteoforms. PFA-fixed MDA-MB-231 cells contained ~15.2% VIM-dimer and 84.8% VIM' proteoform, whereas methanol-fixed MDA-MB-231 cells showed ~50.7% VIM-dimer, 30.0% VIM', and 19.3% VIM" proteoforms. The electrotransfer efficiency, defined as the percentage of the protein target successfully extracted and analyzed by DropBlot, spanned a range from 73.2% to 84.8% in PFA-fixed cells, and a range of 44.4% to 49.3% when in methanol-fixed cells. While DropBlot is suitable for analysis of protein targets from both PFA- and methanol-fixed cells, further optimization may be useful to reduce insoluble protein targets retrieved from methanol-fixed cells.

In addition to heat-induced antigen retrieval explored in this DropBlot report, alternative methods such as enzymatic antigen retrieval (i.e., trypsin, proteinase K, and pepsin) were surveyed in DropBlot. Enzymatic antigen retrieval uses enzymes to digest and break down cross-linked proteins, thus allowing for the recovery of masked protein epitopes that may be inaccessible under standard fixation conditions[72]. We conducted experiments using these enzymatic methods on a PFA-fixed MCF7 cell line and explored various parameters, including enzyme incubation time (5 to 30 min), incubation temperature (room temperature to 37 °C), and lysis temperature (room temperature to 100 °C) (Supplementary Table 4). Through iterative optimization, we identified pepsin as promising in enzymatic antigen retrieval with separation and detection of the VIM and HER2 proteins from the PFA-fixed MCF7 cells. However, weak or undetectable signals were observed for EpCAM and associated proteoforms in all enzymatic antigen-retrieval methods explored here (Supplementary Fig. 16). This lack of signal could be attributed to two factors: (1) damage to the EpCAM surface protein during enzymatic digestion and/or (2) insufficient cell lysis using an antigen-retrieval buffer

composed of 0.5% (w/v) SDS and 6 M urea. Regarding possibility (1): during the enzymatic digestion process, surface proteins, including EpCAM and putative proteoforms, may undergo degradation or structural alterations, leading to a loss or reduction in immunoreactivity[73]. The reduction in immunoreactivity may arise from the enzymatic action, which can disrupt the conformation of epitopes or cause proteolytic degradation of the target proteins. As a consequence, the antibody probes used for detection may not recognize the modified or degraded epitopes, resulting in weak or no signals. Regarding possibility (2): the efficiency of cell lysis is crucial for antigen retrieval methods, as cell lysis efficiency affects the release of target proteins and immunoreactivity. The antigen-retrieval buffer used in our experiments (0.5% (w/v) SDS, 6 M urea), may not complete cell lysis and efficient release of the target proteins. Based on these survey results, enzymatic antigen retrieval is a topic of future study for DropBlot. Incubating cells with urea at elevated temperatures can lead to protein aggregation and carbamylation, leading to adverse impacts on the solubility and immunoreactivity of retrieved antigen targets.

## DropBlot analysis of fixed clinical specimens reports co-expression of two VIM proteoforms in a rare sub-population of HER2+ tumor cells

As a demonstration of the DropBlot assay after assay development and validation, we focused on the analysis of a well-understood protein panel in chemically fixed human-derived biospecimens. As mentioned, the panel consisted of an epithelial marker (EpCAM), a mesenchymal marker exhibiting proteoforms (VIM', VIM"), and human epidermal growth receptor 2 (HER2). In particular, HER2 and VIM were chosen because tumor cells with a mesenchymal phenotype and high expression of HER2 tend to be more aggressive[74,75].

We applied DropBlot to scrutinize single, fixed cells dissociated from 11 solid breast tumor specimens. These human-derived tumor tissues were archived for >6 yrs. stored under −80 °C conditions without chemical fixation. Prior to DropBlot analysis, these patient-derived cells were thawed, tissue was dissociated, and PFA fixation was completed (Fig. 7a, Supplementary Table 5). DropBlot successfully retrieved antigen from 5 of the PFA-fixed cell specimens, as determined by probing for markers of epithelial-to-mesenchymal transition (EMT) and tumor cell growth at the single-cell level (Fig. 7b–c). Having now validated DropBlot on complex human-derived tissue specimens, we sought to understand the sources of VIM-proteoform expression heterogeneity in these specimens, at the level of originating cell type with a particular interest in HER2+ cancer cells (Fig. 7d).

To distinguish cancer cells from among the other cell types present (i.e., stromal, immune), we applied DropBlot in a 'cell gating' mode—analogous to the gating functionality commonly used in flow cytometry. Gating on protein marker expression allows analysis of specific cellular sub-populations in a manner like flow cytometry, with DropBlot enabling analysis of protein proteoforms at the single-cell level, even when antibody probes specific to each proteoform have either poor performance or are nonexistent. This latter functionality is not possible using flow cytometry, or other existing single-cell immunoassays (i.e., immunofluorescence, IHC, mass cytometry).

Employing DropBlot in cell gating mode to hone in on HER2+ cancer cells, we were particularly interested in cancer cells as sources of VIM and VIM-proteoform expression and heterogeneity (Fig. 7e–g). We identified HER2+ cell sub-populations across the five human-derived tumor specimens from which antigen was successfully retrieved after chemical fixation. Attending to Sample #3 in a moment, Samples #1 and #2 were fresh cell suspensions and contained 26.4% (n = 527 cells among 1996 cells analyzed) and 38.9% (n = 474 cells among 1219 cells analyzed) HER2+ cells, respectively. For convenience, Supplementary Table 6 reports the sample size of each starting cell population and each respective subpopulation. While 0% (n = 0 cells, total: 1996) of Sample #1 cells and 7.5% (n = 91 cells, total: 1219) of

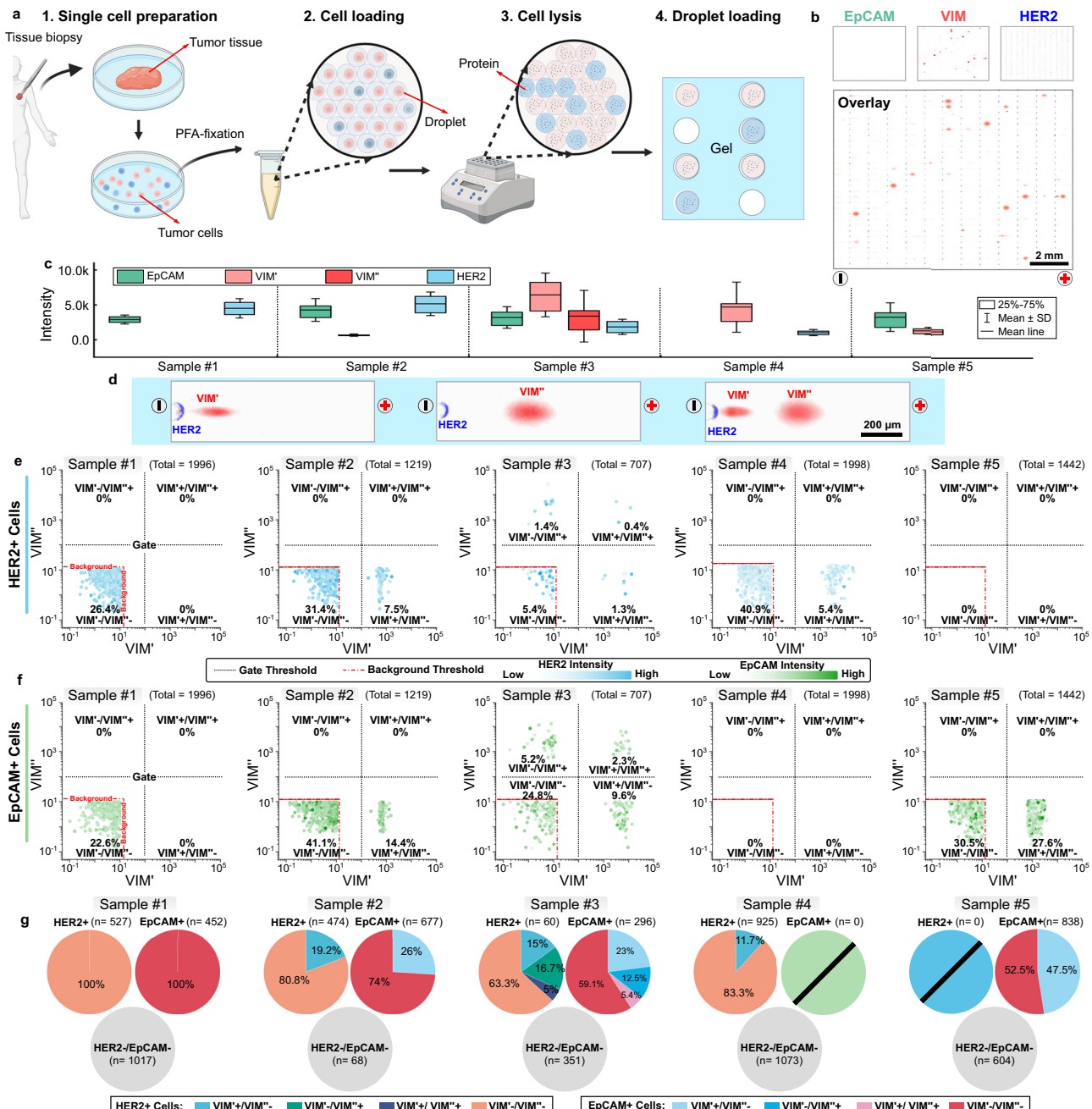

**Fig. 7 | DropBlot analysis of single PFA- and methanol-fixed patient-derived dissociated cancer cells. a** Schematic of clinical sample preparation workflow for DropBlot. PFA conditions: $\Delta t_{fixation} = 15$ min; $\Delta t_{incubation} = 1.0$ h at 98 °C; Created with BioRender.com released under a Creative Commons Attribution-NonCommercial-NoDerivs 4.0 International license. **b** Fluorescence micrographs of single-cell western blots of EpCAM (green), VIM (red), and HER2 (blue) from PFA-fixed tumor cell. Tumor was classified as triple-positive breast cancer. **c** fluorescence intensity of single-cell western blot analyses of PFA-fixed, patient-derived tumor cells for EpCAM, VIM proteoforms (VIM′, VIM″), and HER2. Samples #1–2 were fresh cell suspensions. Sample #3–5 were fresh dissociated tissues. The tumor cells were identified as EpCAM+ or HER2+. On each box plot, the central mark indicates the mean value, the bottom and top edges of the box indicate the interquartile range (IQR), and the whiskers represent the standard deviation (see

Supplementary Table 6 for cell enumeration). **d** Fluorescence micrographs of single-cell western blots of PFA-fixed cells from Sample #3 from **c**, with HER2 (blue) and VIM (red). **e** Cell gating using DropBlot. HER2+ positive cells were further classified based on the expression levels of VIM′ and VIM″. The protein target was considered as negative when the intensity was less than 10. Gate threshold intensity: 100; Background threshold intensity: 10. **f** Cell gating using DropBlot. EpCAM+ positive cells were further classified based on the expression levels of VIM′ and VIM″. The protein target was considered as negative when the intensity was less than 10. Gate threshold intensity: 100; Background threshold intensity: 10. **g** Venn diagram reporting the single-cell target-expression profile for each of single PFA-fixed cells from Sample #1–5 in **c**. See Supplementary Table 6 for cell enumeration of each subpopulation. Source data are provided as a Source Data file.

Sample #2 cells co-expressed VIM′, there were no cells detectable that expressed the VIM″ proteoform (either alone or with VIM). Samples #4 and #5, which were suspensions of fresh dissociated tissues, showed 46.3% (n = 925 cells, total: 1998) and 0% cells as HER2+ (n = 0 cells,

total: 1442), respectively. In Sample #4, 5.4% (n = 108 cells, total: 1998) HER2+ cells co-expressed VIM+, with VIM expression arising only from the VIM′ proteoform. VIM″ was not detected in Sample #4, either alone or co-expressed with VIM′. Sample #5 reported a cell sub-population of

41.9% (n = 604 cells, total: 1442) of the cells analyzed, which was EpCAM-/HER2- and was expressing VIM' alone and likely stromal in origin.

Returning to Sample #3, the 707 cells analyzed were composed of 8.5% HER2+ cells (n = 60 cells, total: 707) by DropBlot. DropBlot detected HER2+ with no co-expression of VIM in ~5.4% (n = 38 cells, total: 707) of the cells assayed. Nearly ~2.7% (n = 19 cells, total: 707) of HER2+ cells expressed one but not the other VIM proteoform (1.3% VIM' only (n = 9 cells, total: 707) and 1.4% VIM" only (n = 10 cells, total: 707)). In addition, DropBlot detected three instances among the analyzed cell population (or 0.4% (n = 3 cells, total: 707)) that expressed HER2+ and also co-expressed both the VIM' and VIM" proteoforms (Fig. 7e, Supplementary Fig. 17).

Importantly, Sample #3 was classified as an invasive ductal carcinoma. Among the HER2+ cell sub-populations surveyed in this pilot study, Sample #3 proved to be the most heterogeneous in VIM proteoform expression. Previous research shows that the mesenchymal phenotype and high expression of HER2 tend to be indicative of a more aggressive phenotype[74,75]. Further, previous studies suggest that the most prevalent form of VIM is the 57-kDa intermediate filament[76], with truncated VIM generated during cancer metastasis[77-79].

Applying DropBlot and gating on sub-populations of non-cancerous cells, we hypothesized that cells expressing VIM only (either form; but no HER2) may represent stromal cells. Gating on total VIM expression, DropBlot reported ~50% of cells analyzed from Sample #3 expressed VIM with co-expression of no other panel marker for this tumor dissociate. Given the prevalence of the VIM-only expressing cells, stromal cells are the likely originating cell type, as would be expected in a dissociated tumor. In the VIM+ stromal cells of Sample #3, 30.8% (n = 218 cells, total: 707) expressed VIM', 8.5% (n = 60 cells, total: 707) expressed VIM", and 10.3% (n = 73 cells, total: 707) co-expressed VIM' and VIM" (Supplementary Fig. 18).

In a function of the DropBlot assay which is gating on differential proteoform expression, we next gated the cell population on expression of just one of the VIM proteoforms (Supplementary Figs. 19 and 20). Just ~1.3% (n = 9 cells, total: 707) of the cells analyzed from Sample #3 co-expressed VIM' with HER2 and ~9.6% (n = 68 cells, total: 707) co-expressed VIM' with EpCAM. Next gating on expression of the second VIM" proteoform only, DropBlot identified ~1.4% (n = 10 cells, total: 707) of the cells analyzed from Sample #3 as co-expressing VIM" with HER2 and ~5.2% (n = 37 cells, total: 707) co-expressing VIM" with EpCAM. Roughly 17.1% (n = 121 cells, total: 707) of the Sample #3 cells analyzed were positive for VIM (VIM' or VIM") and EpCAM, with ~2.3% (n = 16 cells, total: 707) expressing both VIM proteoforms and EpCAM.

Based on these results and demonstrated sufficient performance for PFA- and methanol-fixed cells, we see the potential to further mature and refine DropBlot, including exploring additional sample preparation conditions to determine if additional fixation chemistries are compatible with the sample preparation-to-analysis workflow. While promising for the dual design goals of (i) analysis of sparingly limited cell specimens from biorepositories and (ii) proteoform detection, several areas for obvious performance enhancement and optimization exist, depending on the application area of interest. For example, an immediate next goal could include the integration of established sample preparation techniques, such as empty droplet depletion[80] and single-particle trapping techniques[81], to detect rare cells. With a keen focus on the preparation and analyses of single cells derived from dissociated solid tumor specimens, cells that are not analyzed can be retrieved for archiving and future analysis. A further area entails expanding target-detection multiplexing beyond the handful of targets detected in each cell that is reported in this report. Another area ripe for innovation is in antigen retrieval from formalin-fixed, paraffin-embedded (FFPE) cell specimens. In terms of system design, while DropBlot is utilized here for serial, complementary unit functions (sample preparation to analysis or analysis to detection), we see the modular design as potentially quite powerful for multi-omics questions where coupling of different, complementary analysis stages performed on the same individual cell may lead to breakthroughs in understanding. We also see an opportunity to redesign DropBlot for the handling and analysis of rare cells.

## Methods

### Ethical statement
This research complies with all relevant ethical guidelines. Human tissues used in this study were obtained as de-identified samples from the Stanford Cancer Institute's Tissue Procurement Shared Resource repository and are not considered human subjects as defined by 45 CFR 46, and thus are not regulated by the Office of Human Subjects Protections nor subject to institutional review board (IRB) oversight, as the authors or sponsors have no access to identifying information.

### Statistics and reproducibility
No statistical method was used to predetermine the sample size. Droplet, purified protein, and cell data were collected in technical triplicates to facilitate statistical comparisons between groups. Investigators were not blinded when selecting protein signals with Gaussian distribution. However, data were collected using the 2500-microwell device with randomized treatment methods. The microwells on the same device are technically independent and do not affect each other, ensuring randomization even within a single experiment. All other analyses were automated, effectively making them blind to the investigators.

### Modeling and simulation
Electric field and protein electromigration were simulated using finite-element modeling of electric current and diluted species transport with COMSOL Multiphysics (version 5.5, COMSOL Inc., Sweden). The simulation geometry for the DropBlot PAGE is presented in the top view shown in Fig. 4a. Diffusion coefficients in free solution and within the oil layer were calculated using the Stokes-Einstein equation[82], while the diffusion coefficient in the gel layer was estimated based on published methods[83]. A temperature of 23 °C was used as a baseline in all simulations. The initial concentration of protein in the droplet region was 2 μM, while the initial concentration elsewhere in the system was 0 μM. The model was meshed with a Free Triangle Mesh, and we employed a user-controlled override with a maximum element size of 2 μm and a minimum element size of 0.01 μm in the oil layer regions to provide sufficient mesh density in the narrow region.

### Design and fabrication of droplet-generation chip
Single-emulsion droplet generation chips were designed with Auto-CAD 2021 (Autodesk, San Francisco, CA). SU-8 3050 (Kayaku Advanced Materials, Westborough, MA) was used to fabricate masters with a height of 60 μm following the manufacturer's instructions. PDMS was prepared using a Sylgard 184 Silicone Elastomer Kit (Ellsworth Adhesives, Germantown, WI) and mixed at a ratio of 10:1. After curing (70 °C, 3 hr), the PDMS device was baked for 48 hr at 80 °C to recover hydrophobic characteristics.

### Deterministic droplet-encapsulation of inertially ordered cells
To achieve a high fraction of droplets containing a single mammalian cell, we adopted a technique that deterministically orders cells (into a line) as the cells are approaching the droplet-encapsulation region of the device. Reported by our previous work[84], the method employs sigmoidal microchannels with alternative curvatures. In this stage, the channel Reynolds number was estimated to be 17 with a flow rate of 20 μL/min. The channel design and flow parameters focus cells using inertial lift and Dean drag forces. The cell encapsulation efficiency is affected by gravity-induced changes in the loading density of the cell suspension and will vary over time (Supplementary Fig. 3). We

observed the highest fraction of single-cell containing droplets with a cell-suspension starting concentration of $4.0 \times 10^6$ cells/mL when the loading time is less than 15 min. Noting that the syringe barrel was positioned parallel to gravity, during the minutes-long loading period we observed gravity-induced cell sedimentation, which increases the local concentration of the cell suspension near the syringe outlet, thus affecting the single-cell occupancy rate of the droplets produced[85]. We measured the single-cell droplet occupancy after a 50-min collection period at $12.2 \pm 4.8\%$ (n = 3). Thus, to achieve the highest fraction of single-cell occupancy droplets, we suggest a loading duration of <15 min (considering a 10-min setup period). To reduce or even eliminate the observed gravity-induced cell sedimentation, the use of OptiPrep in the medium will reduce the density difference between the cells and the medium[48].

### Design and fabrication of open-fluidic single-cell western blotting chip

Wafer microfabrication and silanization follow our previous work[39]. Each microwell in the array of 2500 microwells had a diameter of 50 μm and a depth of 60 μm. Microwell-to-microwell spacing was 1000 μm in the X direction and 300 μm in the Y direction. Fabrication of the polyacrylamide gel layer is based on a developed protocol[86]. To prepare an 8%T polyacrylamide gel, the gel precursor solution was mixed with 30% (w/w) acrylamide/bis-acrylamide (Sigma-Aldrich, St. Louis, MO), N-(3-((3-benzoylphenyl) formamido)propyl) methacrylamide (BPMA, PharmAgra Labs, Brevard, NC), 10× Tris-glycine buffer (Sigma-Aldrich, St. Louis, MO) and ddH$_2$O (Sigma-Aldrich, St. Louis, MO). Gels were chemically polymerized for 20 min with 0.08% (w/v) ammonium persulfate (APS, Sigma-Aldrich, St. Louis, MO) and 0.08% (v/v) TEMED (Sigma-Aldrich, St. Louis, MO). After polymerization, gels were carefully released from the wafer by delaminating with a razor blade, and stored in DI water.

### Fluidic generation of W/O droplets for the cell-preparation step

Fresh cells, fixed cells, and purified proteins were resuspended in PBS and injected into the cell inlet of the droplet-generation device (Supplementary Fig. 1). Cell suspension and antigen-retrieval buffer were mixed prior to the emulsion region. 1–2% (v/v) Span® 80 surfactant[87] (Sigma Aldrich, St. Louis, MO) was spiked into mineral oil (Sigma Aldrich, St. Louis, MO) and used as the carrier solution. Flow rates of the substrate and carrier solution were actively controlled by a syringe pump (Chemyx, Stafford, TX). The flow rates of each solution were adjusted to generate droplets of the desired diameter. Droplets were collected in a 1.5 mL Eppendorf® tube or directly loaded onto the top surface of the open-fluidic single-cell western blotting PA-gel using gravity with a customized PDMS droplet-delivery channel and holder (Supplementary Fig. 4).

### Droplet-stability assays for the cell preparation step

By visual inspection, a suspension of stable droplets will maintain two visible layers (top layer: mineral oil, bottom layer: droplets). Once droplet breakage occurs, three layers develop and equilibrate in the collection tube (top layer: mineral oil, middle layer: droplets, bottom layer: antigen-retrieval buffer). Fluorescently labeled protein targets were diluted to 5 μM with PBS, and included Alexa-Fluor 488 labeled immunoglobulin (IgG, Thermo Fisher Scientific, Waltham, MA, A21206), Alexa-Fluor 555 labeled Bovine Serum Albumin (BSA, Thermo Fisher Scientific, Waltham, MA, A34786), Alexa-Fluor labeled Ovalbumin 647 (OVA, Thermo Fisher Scientific, Waltham, MA, O34784), and rTurboGFP (GFP, Evrogen, Russia, FP552). Proteins were encapsulated into 50-μm diameter droplets and observed under an inverted microscope (Olympus IX51, Tokyo, Japan) with a CoolSNAP$_{HQ2}$ camera (Photometrics, Tucson, AZ) for 180 min. Fluorescence images were collected every 3 min, with timing controlled by a mechanical shutter.

### Integration of droplets for the cell preparation step with the single-cell western blot for the cell analysis step

BSA, OVA, and/or IgG proteins were diluted to 0.1 mg/mL and encapsulated in 45-μm droplets containing 0.5% (w/v) SDS. Protein-laden droplets were gravity-settled onto the PA-gel (8%T) surface, which was placed in a customized PAGE chamber (Supplementary Fig. 4, width: 5 cm). A 12.5 mL aliquot of running buffer (1× Tris-glycine and 0.5% (w/v) sodium dodecyl sulfate (SDS, Sigma Aldrich, St. Louis, MO)), was poured onto the chamber. For PAGE, a constant voltage of was applied (200–300 V to obtain $E = 40$–$60$ V/cm) using a DC power supply (Bio-Rad PowerPac Basic, Hercules, CA). At PAGE completion, the applied voltage was set to zero and the protein peaks were photo-captured into the PA gel by applying a 45-s UV exposure (Hamamatsu Lighting cure LC5 UV source, Hamamatsu Photonics, Japan). The chips were rinsed with DI water and imaged with an inverted fluorescence microscope and Genepix® microarray scanner (4300 A, Molecular Devices, San Jose, CA). Exposure time and laser power were held constant for all experiments.

### Culture of cell lines

Human breast cancer cell lines, including MCF7, MCF7/GFP, and MDA-MB-231/GFP were obtained from obtained from the UC Berkeley Tissue Culture Facility and were cultured in DMEM medium (Thermo Fisher Scientific, Waltham, MA); One human lung cancer cell line (H1299, ATCC, Manassas, VA, CRL-5803) and fibroblast cell line (ATCC, Manassas, VA, PCS-201-012) were cultured in RPMI medium (Thermo Fisher Scientific, Waltham, MA). Both mediums were supplemented with 10% (v/v) fetal bovine serum (GeminiBio, West Sacramento, CA), 1% (v/v) penicillin/streptomycin solution (Thermo Fisher Scientific, Waltham, MA), and 0.1 mM non-essential amino acid solution (Thermo Fisher Scientific, Waltham, MA) in an incubator (37 °C, 5% CO2). Prior to DropBlot analysis, adherent cells were released through incubation with 0.05% trypsin-EDTA solution (Thermo Fisher Scientific, Waltham, MA). The concentration of harvested cells was measured with a hemocytometer (Hausser Scientific, Horsham, PA) and resuspended with PBS (Thermo Fisher Scientific, Waltham, MA) to the desired concentration. Short tandem repeat (STR) profiling has been conducted for all the cell lines by the UC Berkeley Tissue Culture Facility or ATCC. Mycoplasma contamination test was conducted annually by the UC Berkeley Tissue Culture Facility, and all cell lines were tested negative.

### Preparation of peripheral blood mononuclear cells (PBMCs)

The human buffy coat blood was procured from a commercial supplier (Zen-Bio, Research Triangle, NC, SER-WB-SDS) and mixed with an equal volume of non-complemented DMEM medium. Peripheral blood mononuclear cells (PBMCs) were isolated from this blood sample using a gradient separation method. Specifically, 20 mL of Ficoll-Paque™ PLUS (Cytiva, Marlborough, MA) was carefully layered at the bottom of a 50 mL conical tube, followed by the addition of 30 mL of the diluted blood. The tube was then centrifuged at room temperature at 760 g for 20 minutes without applying brakes. Subsequently, PBMCs were collected from the interface between the Ficoll and plasma layers and washed thrice with PBS through centrifugation at $350 \times g$ for 8 minutes each time. The concentration of harvested cells was measured with a hemocytometer (Hausser Scientific, Horsham, PA) and resuspended with PBS (Thermo Fisher Scientific, Waltham, MA) to the desired concentration.

### Application of DropBlot to fresh and fixed cells

The cells MCF7, MDA-MB-231, H1299, and PBMCs were resuspended with PBS to a concentration of $4 \times 10^6$ cells/mL. The antigen-retrieval buffer used for live-cell lysis was 2× Tris-glycine buffer (Sigma Aldrich, St. Louis, MO) supplemented with 1% (w/v) SDS and 12 M urea (Sigma Aldrich, St. Louis, MO). Flow rates of cell solution, antigen-retrieval

buffer, and carrier layer (oil) were 1.0, 1.0, 12.0 μL/min to generate droplets with diameters of ~45 μm. The final concentration of antigen-retrieval buffer was 1× Tris-glycine supplemented with 0.5% (w/v) SDS and 6 M urea. Cell-laden droplets were loaded onto the top surface of the single-cell western blotting device (8%T polyacrylamide gel, stippled with microwells). The PDMS slab was then relocated to cover the droplet-filled microwells and excess mineral oil was flushed out with running buffer (1× Tris-glycine supplemented with 1% (w/v) SDS). Cells were lysed for at least 10 min once encapsulated in droplets at room temperature. 12.5 mL of running buffer was poured into the chamber and protein target electro-transfer and PAGE were initiated by applying a 300-V constant voltage for 30 s across the device to achieve an average $E = 40$–$60$ V/cm across the gel. At PAGE completion, the applied potential was zeroed out, the proteins halted electromigration, and stationary protein peaks were photo-captured to the benzophenone in the PA gel by application of a 45 s pulse of UV light. The gel was then rinsed briefly with deionized water and stored in Tris-buffered saline with Tween 20 (TBST, Cell Signaling Technology, Danvers, MA) overnight to remove excess oil and residual antigen-retrieval buffer.

To generate PFA-fixed cells, the cancer cell lines MCF7 and MDA-MB-231 were fixed with 4% paraformaldehyde (PFA, Alfa Aesar, Haverhill, MA) for 15–30 min at room temperature following the manufacturer's protocol. After fixation, cells were washed 3× with PBS to remove excess PFA and resuspended with PBS to achieve a concentration of ~4 × 10^6 cells/mL. The antigen-retrieval buffer used for live-cell PFA fixation was 2× Tris-glycine buffer supplemented with 1% (w/v) SDS and 12 M urea. Flow rates of cell solution, antigen-retrieval buffer, and carrier layer (oil) were 0.5, 0.5, and 5 μL/min, respectively, to generate 45-μm diameter droplets. The final concentration of antigen-retrieval buffer was 1× Tris-glycine buffer supplemented with 0.5% (w/v) SDS and 6 M urea. The droplets were collected and incubated in a 1.5 mL Eppendorf tube at 98 °C for 1–2 hr, and then loaded onto the 8%T polyacrylamide gel. Running buffer (12.5 mL) was poured into the chamber and the separation was initiated by supplying 300-V constant voltage for an average electric field of 40–60 V/cm across the gel for 30 s. After PAGE, the proteins were photo-captured by 45-s UV exposure. The gel was then rinsed briefly with deionized water and stored in TBST.

For methanol-fixed cells, the cancer cell lines MCF7 and MDA-MB-231 were again used as model cells, now fixed with ice-cold methanol (Sigma Aldrich, St. Louis, MO) for 15 min at −20 °C following the manufacturer's protocol. After fixation, cells were washed 3× with PBS to remove excess methanol and resuspended with PBS to achieve a concentration of ~4 × 10^6 cells/mL. The live-cell antigen-retrieval buffer was 2× Tris-glycine buffer supplemented with 1% (w/v) SDS and 12 M urea. Flow rates of the cell solution, antigen-retrieval buffer, and carrier layer (oil) were 0.5, 0.5, and 5 μL/min, respectively, to generate 45-μm diameter droplets. The final concentration of antigen-retrieval buffer was 1× Tris-glycine buffer supplemented with 0.5% (w/v) SDS and 6 M urea. The droplets were collected and incubated in a 1.5 mL Eppendorf® tube at 98 °C for 1 hour, and then loaded onto 8%T polyacrylamide gel. 12.5 mL of running buffer was poured into the chamber and the separation was initiated by supplying 300 V constant voltage for 30–120 s to reach an average electric field of 40–60 V/cm across the gel. After PAGE, the proteins were photo-captured by applying UV light to the gel for 45 s. The gel was then rinsed briefly with deionized water and stored in TBST.

## Antigen retrieval using enzymatic methods

The cancer cell line, MCF7, was fixed with PFA for 15 min at room temperature. The fixed cells were washed with PBS three times and aliquoted into a 1.5 mL Eppendorf tube with 1 × 10^6 cells/tube. Trypsin antigen retrieval solution (ab970, Abcam, Cambridge, United Kingdom), proteinase K antigen retrieval solution (ab64220), and pepsin

antigen retrieval solution (ab64201) were used following the manufacturer's instructions. The detailed incubation and lysis conditions are listed in Supplementary Table 4. After incubation, cells were encapsulated in 45-μm diameter droplets and the protein lysate was separated by supplying 300-V constant voltage across the gel for 30 s. After PAGE, the proteins were photo-captured by 45-s UV exposure. The gel was then rinsed briefly with deionized water and stored in TBST.

## Patient-derived tissue samples from a biospecimen repository

Primary-tumor tissue was obtained anonymized and blinded from Stanford Cancer Institute's Tissue Procurement Shared Resource facility. Human-derived tumor specimens were archived for >6 yr prior to DropBlot analysis, stored at −80 °C.

## Fresh tumor specimens (cell suspension or tissue)

Frozen samples were thawed in a water bath at 37 °C for 1 min and mixed with 10 mL of DMEM medium. Samples were centrifuged at $300 \times g$ for 5 min to remove the supernatant. For the fresh cell suspension, the samples were fixed with 4% PFA for 15 min and resuspended with PBS to a concentration of 4 × 10^6 cells/mL. For the fresh tissue (Fig. 7a), a 1-g tissue specimen was weighed and placed in a petri dish containing 5 mL of 37 °C DMEM medium. Using a scalpel and tweezer, the tissue was coarsely dissected into fragments <0.75 mm in diameter. A tissue suspension was constituted by adding 5 mL of Tumor & Tissue Dissociation Regent (TTDR, BD Bioscience, San Jose, CA), and then incubating the mixture at 37 °C for 30 min with frequent agitation. After incubation, 25 mL of Dulbecco's Phosphate Buffered Saline (DPBS, Thermo Fisher Scientific, Waltham, MA) containing 1% BSA and 2 mM EDTA (Thermo Fisher Scientific, Waltham, MA) was added. Large tissue/cell clusters were removed with a 70-μm cell strainer and then centrifuged at $300 \times g$ for 5 min to remove the supernatant. A cell pellet formed and was resuspended in 2 mL of 1× lysis buffer (Tonbo Biosciences, San Diego, CA) and incubated at room temperature for 15 min. A 40 mL aliquot of DPBS containing 1% BSA and 2 mM EDTA was then added to the mixture. After removing the supernatant, the cells were fixed following the fixation protocols described elsewhere. Cells were counted using a hemocytometer and resuspended to 4 × 10^6 cells/mL with PBS.

## Formalin-fixed, paraffin-embedded (FFPE) tumor specimens

Frozen tissue samples were thawed in a water bath at 60 °C for 2 hr, and bathed in 10 mL xylene (Sigma Aldrich, St. Louis, MO) for 5 min (twice). The samples were rehydrated with 96% ethanol, 90% ethanol, 70% ethanol, 50% ethanol, and PBS for 5 min, then washed twice. The cells were fixed following the fixation protocols described elsewhere and resuspended to 4 × 10^6 cells/mL with PBS.

## Immunoprobing and fluorescence imaging for the cell analysis step

The primary antibody immunoprobing solution was prepared by diluting stock solutions of primary antibodies in 2% (w/v) BSA/TBST solution to achieve an antibody concentration of 0.05 μg/μL (single antibody). Primary antibodies used were EpCAM (Abcam, AB32392), VIM (Abcam, AB8978, AB92547) HER2 (Abcam, AB16901), GAPDH (Sigma-Aldrich, SAB2500450, AB_10603419), FAP (Abcam, ab207178), and CD45 (Abcam, AB8216). The single-cell western blotting device (gel slide) was treated with 80 μL of primary antibody immunoprobing solution and incubated at room temperature for 2 hr. After incubation, each single-cell western blotting device was washed twice with TBST buffer for 1 hour. The secondary antibody immunoprobing solution was prepared by diluting stock solutions of primary antibodies in 2% (w/v) BSA/TBST solution to achieve a concentration of 0.05 μg/μL (single antibody). Secondary antibodies used in this project include Alexa Fluor 488 donkey anti-rabbit (Thermo Fisher Scientific, cat. no. A21206, RRID: AB_2535792), Alexa Fluor 555 donkey anti-goat (Thermo

Fisher Scientific, cat. no. A21432, RRID: AB_2535853), Alexa Fluor 594 donkey anti-mouse (Thermo Fisher Scientific, cat. no. A21203, RRID: AB_141633), Alexa Fluor 647 donkey anti-mouse (Thermo Fisher Scientific, cat. no. A32787, RRID: AB_2762830). The single-cell western blotting device was incubated with 80 μL of secondary antibody immunoprobing solution at room temperature for 2 hr. After incubation, the single-cell western blotting device was washed twice with TBST buffer for 1 hr. Before fluorescence imaging, the single-cell western blotting device was washed 3× with DI water to remove excess salts, and dried with nitrogen gun. The single-cell western blotting device was imaged with a Genepix Microarray Scanner. Images were analyzed using custom analysis scripts in MATLAB (MathWorks, Natick, MA). Two to three protein targets were analyzed concurrently. Antigen-target multiplexing utilized an established stripping and reprobing method[86]. Briefly, the gel slide was treated with stripping buffer (0.8%(v/v) beta-mercaptoethanol, 2%(w/v) SDS, 62.5 mM Tris-HCl, in ddH$_2$O) at 55 °C for 30 min to remove probing antibodies. After stripping treatment, the slide is ready for the second round of probing with new antibodies.

### Reporting summary
Further information on research design is available in the Nature Portfolio Reporting Summary linked to this article.

## Data availability
Data related to the figures has been deposited in the Figshare database (https://doi.org/10.6084/m9.figshare.25335568)[88]. Source data are provided with this paper.

## Code availability
All custom simulation code and data analysis code generated as a part of this work are accessible via GitHub (https://github.com/liulabUGA/DropBlot_Code/) and zenodo (https://doi.org/10.5281/zenodo.11437115).

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

## Acknowledgements
This work was funded by the National Institutes of Health (NIH) R01CA20301 (A.E.H.). We sincerely thank Herr Lab members Gabriela Lomeli and Ana Gomez Martinez for initial training on the operation of the single-cell western blot. We acknowledge all members of the Herr Lab at UC Berkeley. We are grateful to the R&D Machine Shop at UC Berkeley for the fabrication of the chamber and to patients who selflessly donated biospecimens to the Stanford Cancer Institute's Tissue Procurement Shared Resource facility. We thank Dr. Leidong Mao and Dr. Yao Yao from the University of Georgia for kindly sharing cancer cell lines, PBMCs, and fibroblasts.

## Author contributions
Y.L. and A.E.H. designed the hybrid microfluidic platform (DropBlot). Y.L. and A.E.H. designed experiments. Y.L. performed DropBlot assay on purified proteins, fresh & fixed cell lines, and fresh & fixed clinical samples. Y.L. designed software and performed data analysis. Y.L. and A.E.H. wrote the manuscript.

## Competing interests
The authors declare no competing interests.
