## [Peer Review File · Nature Communications]

REVIEWER COMMENTS

Reviewer #1 (Remarks to the Author):

This manuscript describes development of a sample preparation system based on droplets for a western-blot type analysis that uses the gel array system previously developed and reported by the Herr lab. The system is shown to allow protein sample preparation from fresh and fixed single cells with blots run in parallel. This is a cool system that advances the state of the art for single cell analysis. I think the goals of performing multiplexed antibody based analysis on single cells in parallel, including from fixed cells, is a highly worthy one that will interest many reviewers. The manuscript is thorough walking through several optimization and development steps with regard to droplets and sample preparation. Overall the paper is exciting and extremely well written.

I have a few comments that the authors will hopefully address.

- 1) When isolating single cells the authors found that over 70% of the droplets contained single cells. Typically when isolating single cells into droplets Poisson distributions are expected and this strikes me as a high percentage of single cells for such a distribution. Can the authors comment on if their results follow such distributions and if not how did they overcome it?
- 2) The manuscript points out that after fixation, the resulting migration times are 50% slower compared to from fresh cells. Often in a western blot, one compares to a ladder to confirm roughly the MW. Even without this, it would seem that this migration does impact the ability to confidently call identifications based on just the antibody response. (Antibodies are of course very good, but cross-reactivity and non specific effects and multiple isoforms are all reasons that the electrophoretic separation is useful.)
- 3) Related to the above comment, when discussing the effect of fixation on migration time the paper says “as makes sense given what is known about the PFA fixation mechanism”. Could they provide a short elaboration there, specifically is this due to the protein migrating as a modified (heavier) species or some other effect?
- 4) My interpretation of the discussion on relative amounts of fixed cells is that relative quantification in fixed cells remains an unsolved problem. Is that a fair conclusion? Perhaps discuss this current limitation a bit more clearly if so.
- 5) This is a small issue, but Figure 4 refers to a protein “ladder”, but only see 2 compounds at a time. Perhaps it is a bit much to call it a ladder.

Reviewer #2 (Remarks to the Author):

The manuscript builds on the landmark single-cell western blotting paper from the same group published in 2014 in Nature Methods. In this contribution they extend the scope of the technique to fixed cells to make use of the enormous back catalogue of valuable clinical samples.

The concept involves first encapsulating single cells in microfluidic droplets along with reagents for cell lysis and antigen retrieval. Droplets containing cell lysate with antigens retrieved from fixative are subsequently parked in an array of ~5000 microfabricated wells, followed by electrophoresis, capture and immunoprobing. The manuscript provides a detailed account of how the assay was developed, documenting the many considerations and obstacles needing to be solved to meet the challenges of the new assay. Droplets remain intact and without cross-talk despite the harsh chemical environment required for antigen retrieval. Cell lysates are retrieved from droplets with electrophoretic migration in the gel with immunostaining to measure distribution of key antigens (EpCAM, VIM, HER2, GAPDH between cells). The method was successfully developed using fixed cell lines and also >6-year-old clinical samples. The authors have yet to refine the method for formalin-fixed paraffin embedded (FFPE) tissue samples.

The manuscript raises conceptual and practical concerns. Using a droplet environment for antigen retrieval limits throughput. The microwell platform has a real-estate containing 5,000 electrophoresis lanes. However, to achieve single-cell resolution, most droplets will be empty, with the authors reporting the analysis of 100's of single cells per run (wells contained 79.5% single cells, implying multi-cell occupancy in 20.5% cells). The scale of the assay is not sufficient to effectively survey cell heterogeneity in a tissue section. In contrast, standard cytometry achieves kHz processing, making use of all available cells in the tissue section. The droplet format has a further disadvantage in that washing steps cannot be used - the use of 6M urea produces multiple peaks per protein is one point of concern. Also, the multiplexing capability given available space and multi-peak separations is limited and will be rivalled by cytometry (up to 20-plex) or mass cytometry (up to 40-plex). One merit of single-cell blotting is the ability to identify isoforms with a single antibody). However, antibodies specific to different isoforms (of research and clinical value) are commercially available.

In summary, the manuscript documents good progress in tackling the different challenges of processing fixed cells in droplets and interfacing this with a single-cell electrophoresis array. Despite the good progress, the method and overall concept is unlikely to replace existing approaches and make an impact in the field which places increasing demands on throughput and multiplexing.

Please note occasional typos were spotted.

Reviewer #3 (Remarks to the Author):

The authors have developed a method to perform single-cell western blots. I will specifically comment on the application of this method to breast cancer.

In figures 5&6, the authors demonstrate the expression of EPCAM, Vimentin and Her2 in unfixed and fixed cells, respectively. These are relevant markers for breast cancer cell lines as they are biologically relevant and often heterogeneously expressed. These figures appear to technically validate the method but do not reveal new biological insights.

In Figure 7, the authors apply this same panel of antibodies to cryopreserved clinical samples. While they observe heterogenous expression, this is well documented previously using methods such as immunofluorescence. Two of these markers, Epcam and Vim, are not of any clinical value.

There are many methods to make single cell protein measurements in clinical samples, using methods based on multiplexed immunofluorescence, that can scale up to 100 antigens. It seems that the specific advantage of this method would be in identifying different molecular weight forms of proteins, but the authors have not chosen targets where that information is relevant or important and have not followed up on the relevance of the 2 Vimentin forms reported.

I am concerned that the authors have taken whole tumour dissociates, which will contain not just cancer cells but also stromal cells (which are mostly vimentin+) and immune cells (which are mostly negative for all markers). Therefore the reported observation of heterogenous expression of proteins is likely to be a product of varying cellular composition, rather than heterogenous expression by cancer cells as the authors conclude. For instance, the 2 vimentin isoforms may be expressed by epithelial vs stromal cells. The authors do not validate their findings.

Response to Reviewer 1

We thank the Reviewer for their valuable comments. We have endeavored to fully and clearly address each concern (outlined in detail below) through modifications and additions to the original submission.

Reviewer's preface comment: *This manuscript describes development of a sample preparation system based on droplets for a western-blot type analysis that uses the gel array system previously developed and reported by the Herr lab. The system is shown to allow protein sample preparation from fresh and fixed single cells with blots run in parallel. This is a cool system that advances the state of the art for single cell analysis. I think the goals of performing multiplexed antibody based analysis on single cells in parallel, including from fixed cells, is a highly worthy one that will interest many reviewers. The manuscript is thorough walking through several optimization and development steps with regard to droplets and sample preparation. Overall the paper is exciting and extremely well written.*

Authors' preface comment response: We thank the Reviewer for their enthusiasm regarding the advent of new single-cell tools suitable for protein analyses of fixed cells, an underserved area with long-term importance to clinical medicine.

Reviewer comment 1: *I have a few comments that the authors will hopefully address. When isolating single cells the authors found that over 70% of the droplets contained single cells. Typically when isolating single cells into droplets Poisson distributions are expected and this strikes me as a high percentage of single cells for such a distribution. Can the authors comment on if their results follow such distributions and if not how did they overcome it?*

Authors' response 1: We agree and thank the Reviewer for suggesting that we elaborate on the approach used to generate a high fraction of droplets containing single cells. We expand the description of our adoption of a published technique for deterministic droplet-encapsulation of inertially ordered cells. The comment echoes a suggestion from Reviewer #2 (below). We make the following revisions to the **Results & Discussion** section and add a new **Supplemental Figure**, as follows:

Page 7 of the **Results & Discussion** section now reads: "Third, we sought a stable-droplet formulation robust to mechanical handling and seating of said droplets into the open-fluidic array of microwells. To achieve a high fraction of droplets that contain a single cell, we adopted a technique for deterministic droplet-encapsulation of inertially ordered cells³⁹."

Page 8 of the **Results & Discussion** section now reads: "Our first step was systematically optimizing the droplet generation function for single cell per droplet occupancy, by considering channel geometry, flow rates of the continuous and dispersed phases, and the initial concentration of the cell suspension (MCF7 cells, **Figure S1**). To ensure a high fraction of droplets contained a single cell, we designed the droplet-loading chip to order cells using a spiral geometry that achieves inertial focusing of the cells for subsequent ordered loading of droplets (**Figure S2**), as has been reported recently^{39,40}. Single-cell occupancy was achieved in $79.5 \pm 6.1\%$ ($n = 3$) of droplets generated under the following empirically determined conditions: droplet diameter = $\varnothing_{droplet} = 40\text{-}60 \mu\text{m}$; volumetric flow rate of dispersed phase = $Q_{dispersed} = 0.2\text{-}20 \mu\text{L}/\text{min}$; volumetric flow rate of continuous phase = $Q_{continuous} = 0.2\text{-}100 \mu\text{L}/\text{min}$; starting concentration of cell suspension = $1.0\text{-}6.0 \times 10^6 \text{ cells}/\text{mL}$. The highest fraction of single-cell containing droplets was achieved when the starting concentration of the cell suspension was $4.0 \times 10^6 \text{ cells}/\text{mL}$."

Addition of new works cited:

(39) Moon, H.-S.; Je, K.; Min, J.-W.; Park, D.; Han, K.-Y.; Shin, S.-H.; Park, W.-Y.; Yoo, C. E.; Kim, S.-H. Inertial-ordering-assisted droplet microfluidics for high-throughput single-cell RNA-sequencing. *Lab on a Chip* **2018**, *18* (5), 775-784.

(40) Tang, T.; Zhao, H.; Shen, S.; Yang, L.; Lim, C. T. Enhancing single-cell encapsulation in droplet microfluidics with fine-tunable on-chip sample enrichment. *Microsyst Nanoeng* 2024, 10, 3. DOI: 10.1038/s41378-023-00631-y

Page 24 of the **Materials & Methods** section now reads: “**Deterministic droplet-encapsulation of inertially ordered cells.** To achieve a high fraction of droplets containing a single mammalian cell, we adopted a technique that deterministically orders cells (into a line) as the cells are approaching the droplet-encapsulation region of the device. Reported by our previous work⁷⁶, the method employs sigmoidal microchannels with alternating curvatures. In this stage, the channel Reynolds number was estimated to be 17 with a flow rate of 20 $\mu\text{L}/\text{min}$. The channel design and flow parameters focus cells using inertial lift and Dean drag forces.”

The **Supplemental Information** section now includes a new **Figure S2** which reports the inertial-focusing chip design accompanied by fluorescence micrographs of the deterministic cell-ordering function:

Figure S2. Deterministic droplet-encapsulation of inertially ordered cells. (Top) CAD schematic of droplet-generation chip designed to utilize inertial focusing conditions to order single cells in a ‘single-file queue’ prior to loading cells into droplets at a target occupancy of one cell per droplet. (Bottom) Fluorescence micrographs from chip regions demarcated with green boxes (in top CAD schematic) show inertial focusing of a stream of cells expressing red fluorescence signal (left) and the downstream, ordered flow of cells prior to encapsulation in droplets (right).

Reviewer comment 2: *The manuscript points out that after fixation, the resulting migration times are 50% slower compared to from [sic] fresh cells. Often in a western blot, one compares to a ladder to confirm roughly the MW. Even without this, it would seem that this migration does impact the ability to confidently call identifications based on just the antibody response. (Antibodies are of course very good, but cross-reactivity and non specific effects and multiple isoforms are all reasons that the electrophoretic separation is useful.)*

Response to comment 2: We agree and revise the manuscript to more fully report on the well-accepted assay development approach we adopted (*The Assay Guidance Manual*, NCATS NIH, 2004-now) to account for biospecimen matrix effects anticipated in – and observed in – the chemically complex fixed-cell specimens. Our revisions include:

Page 6 of the **Introduction** now reads: “We develop the DropBlot workflow using two breast-cancer cell lines and a suite of chemical fixation conditions. Each protein target is first individually identified by immunoreactivity from fresh-cell (unfixed) lysate and then under each fixation condition for a well-characterized pair of cancer cell lines. We expect each protein target to vary in electromigration behavior

depending on the fresh or fixed state of the originating cell, thus methodical assay development and target-identity tracking are utilized. After validating the identity and immunoreactivity of each protein target, we next apply DropBlot to investigate a pilot cancer-protein panel composed of an epithelial marker (EpCAM), a mesenchymal marker (vimentin, VIM), and human epidermal growth receptor 2 (HER2) in a pilot group of breast cancer patient-derived cell samples.”

Page 14 of the **Results & Discussion** now reads: “To assess the relationship between antigen-retrieval buffer formulation (including SDS) and the immunoreactivity of any retrieved antigen, we scrutinized fresh cells from well-studied cultured cell lines and the protein targets epithelial cellular adhesion molecule (EpCAM, 35-40 kDa) and intermediate filament protein (VIM, 57 kDa). We anticipate the chemical state of each originating cell will impact the degree of antigen retrieval for each protein target. Antigen retrieval means, most obviously, the amount of antigen material recovered from a chemically fixed cell. Importantly, and more subtly, antigen retrieval also depends on the degree of recovery of immunoreactivity, which in turn depends on the physicochemical properties of each retrieved antigen species. In this latter aspect, we anticipate that fixation-induced alterations in protein physicochemical properties will inherently affect electromigration. Consequently, we start by establishing protein-target identity in the simplest to the most complex matrix conditions using a modified ‘spiked recovery’ immunoassay development process to account – as much as possible – for matrix effects anticipated in human-derived and chemically fixed cell specimens⁵¹. Here, that means we start with target antigen measurement from a clear buffer, progressing next to antigen measurement from fresh cell lines, then retrieval from fixed cells from the same cell lines, and finally move to considering retrieval of a panel of protein targets in fixed patient-derived tumor specimens.”

Addition of new works cited:

(51) Cox, K. L.; Devanarayan, V.; Kriauciunas, A.; Manetta, J.; Montrose, C.; Sittampalam, S. Immunoassay methods. *Assay Guidance Manual [Internet]* 2019.

Page 16 of the **Results & Discussion** now reads: “... DropBlot successfully completed unfixed-cell and protein-sample preparation, then supported immunoblotting of the four targets as shown in **Figure 5c-d**. We observed no cross-reactivity between the antibody probes utilized and the four targets from the cancer-protein panel.”

Page 18 of the **Results & Discussion** now reads: “Similarly, we scrutinized methanol-fixed MCF7 cells and observed that, while EpCAM was detectable by single-cell immunoblotting, electromigration was even slower than the PFA-fixed cells (**Error! Reference source not found.f**). We further observed elevated protein signals near the microwells (**Error! Reference source not found.g-h, Figure S14**), suggesting the presence of large protein dimer molecules or even protein aggregates. Recall, we anticipate that the physicochemical properties of each retrieved antigen (including electromigration, degree of immunoreactivity, and amount of protein material solubilized) will differ depending on the antigen and the chemical conditions of the originating cell, thus the use of a methodical assay development process that starts with fresh-cell lysate from well-characterized cell lines, then assays a suite of chemical fixation conditions for those same cell lines and protein targets, before moving to target identification in chemically fixed, human-derived biospecimens⁵¹.”

Page 20, **Results & Discussion** now reads: “**DropBlot analysis of fixed clinical specimen reports co-expression of two VIM proteoforms in a rare sub-population of HER2+ tumor cells**. As a first demonstration of the DropBlot assay after assay development and validation, we focused on analysis of a well-understood protein panel in chemically fixed human-derived biospecimens. As mentioned, the panel consisted of an epithelial marker (EpCAM), a mesenchymal marker exhibiting proteoforms (VIM’, VIM’'), and human epidermal growth receptor 2 (HER2). In particular, HER2 and VIM were chosen because tumor cells with a mesenchymal phenotype and high expression of HER2 tend to be more aggressive^{69, 70}.”

Reviewer comment 3: Related to the above comment, when discussing the effect of fixation on migration time the paper says “as makes sense given what is known about the PFA fixation mechanism”. Could they provide a short elaboration there, specifically is this due to the protein migrating as a modified (heavier) species or some other effect?

Authors' response 3: We thank the Reviewer and revise the **Results & Discussion Section** and add **Figure S12** to elaborate on the PFA fixation mechanism with additional **references**.

Page 17, **Results & Discussion** now reads: “During PAGE, we further observed protein electromigration as being ~50% slower for targets retrieved from PFA-fixed cells as compared to the respective fresh cancer cell lines (**Figure 6a-b**). Reduced electromigration of PFA-fixed cells versus fresh cells is expected because the chemical-fixation chemistry is known to form an insoluble network of molecules within cells by creating new covalent molecular bonds⁶³. Based on this PFA-fixation mechanism, PFA-fixed protein species would reasonably have a larger Stokes radius and concomitantly lower electrophoretic mobility (**Figure S12**), as compared to unfixed (denatured) protein targets.”

With the **Supplementary Information** revised to include new data and analysis, as referenced in the above revision to the **Results & Discussion** section:

Figure S12. Calibration of electromigration for endogenous antigen targets retrieved from PFA-fixed cells using a 3-component soluble protein ladder. (a) Fluorescence micrographs of antigen targets analyzed by the western blotting assay used for DropBot. (Top panel) Fluorescence micrograph of PAGE electromigration for a soluble 3-component protein ladder composed of: AF555-BSA (pink), AF647-OVA (blue), and AF488-IgG (cyan). (Bottom panel) Fluorescence micrographs of single-cell western blotting by DropBlot for two endogenous antigen targets retrieved from PFA-fixed MDA-MB-231 cells ($\Delta t_{\text{fixation}} = 15 \text{ s}$, $\Delta t_{\text{incubation}} = 1.0$ at 98°C). (b) Molecular-mass calibration of two endogenous antigen targets retrieved from PFA-fixed cells shows impact of PFA-chemical fixation on physicochemical properties of protein molecules. Once retrieved from PFA-fixed cells, endogenous antigen target EpCAM spuriously (but not unexpectedly) electromigrates as a 120 kDa protein (versus known molecular mass of 30-40 kDa) and endogenous VIM spuriously (but not unexpectedly) electromigrates as a 170 kDa protein (versus known molecular mass of 57 kDa). Conditions: $\Delta t_{\text{PAGE}} = 30 \text{ s}$; $E = 40 \text{ V/cm}$.

Reviewer comment 4: My interpretation of the discussion on relative amounts of fixed cells is that relative quantification in fixed cells remains an unsolved problem. Is that a fair conclusion? Perhaps discuss this current limitation a bit more clearly if so.

Authors' response 4: Thank you for the excellent suggestion to elaborate on the limitation of assessing relative protein expression, in light of the novel ‘per cell’ aspect of this single-cell western blot.

Page 5 of the **Introduction** now reads: “Slab-gel immunoblotting of fixed samples has been reported^{7, 8}, often pooling tissue or cell samples to enhance detection sensitivity. Our lab introduced a suite of immunoblotting tools optimized for individual unfixed (fresh) cells^{9, 10}, including for analysis of human-derived dissociated solid tumor¹¹ and circulating tumor cell specimens¹². For fresh or fresh-frozen clinical specimens, our research group has introduced single-cell immunoblotting^{9, 10}. In addition to obtaining

cellular- and subcellular-resolution protein profiles, precision immunoblotting allows researchers to more directly and quantitatively compare cell-to-cell levels of protein expression. Since expression of each protein target is measured for each microwell-isolated cell, expression can be readily normalized on a per-cell or even per cell-volume basis.”

Reviewer comment 5: *This is a small issue, but Figure 4 refers to a protein “ladder”, but only see 2 compounds at a time. Perhaps it is a bit much to call it a ladder.*

Authors' response 5: We agree and apologize for the inaccuracy. We revise the **Figure 4** caption and the **Main Text** to reference “protein standards” removing all instances of the term “ladder”.

Response to Reviewer #2

Reviewer preface comment: *The manuscript builds on the land mark single-cell western blotting paper from the same group published in 2014 in Nature Methods. In this contribution they extend the scope of the technique to fixed cells to make use of the enormous back catalogue of valuable clinical samples. The concept involves first encapsulating single cells in microfluidic droplets along with reagents for cell lysis and antigen retrieval. Droplets containing cell lysate with antigens retrieved from fixative are subsequently parked in an array of ~5000 microfabricated wells, followed by electrophoresis, capture and immunoprobing. The manuscript provides a details account of how the assay was developed, documenting the many considerations and obstacles needing to be solved to meet the challenges of the new assay. Droplets remain intact and without cross-talk despite the harsh chemical environment required for antigen retrieval. Cell lysates are retrieved from droplets with electrophoretic migration in the gel with immunostaining to measure distribution of key antigens (EpCAM, VIM, HER2, GAPDH between cells). The method was successfully developed using fixed cell lines and also >6-year-old clinical samples. The authors have yet to refine the method for formalin-fixed paraffin embedded (FFPE) tissue samples.*

Reviewer comment 1: *The manuscript raises conceptual and practical concerns. Using a droplet environment for antigen retrieval limits throughput. The microwell platform has a real-estate containing 5,000 electrophoresis lanes. However, to achieve single-cell resolution, most droplets will be empty, with the authors reporting the analysis of 100's of single cells per run (wells contained 79.5% single cells, implying multi-cell occupancy in 20.5% cells).*

Authors' response 1: Similar to Reviewer 1's comment 4 (above), we thank this Reviewer and elaborate on our adoption of a technique for deterministic droplet-encapsulation of inertially ordered cells prior to droplet loading with cells. We revise the **Results & Discussion** section and add a new **Figure S2**, as follows:

Page 7 of the **Results & Discussion** section now reads: “Third, we sought a stable-droplet formulation robust to mechanical handling and seating of said droplets into the open-fluidic array of microwells. To achieve a high fraction of droplets that contain a single cell, we adopted a technique for deterministic droplet-encapsulation of inertially ordered cells³⁹.”

Page 8 of the **Results & Discussion** section now reads: “Our first step was systematically optimizing the droplet generation function for single cell per droplet occupancy, by considering channel geometry, flow rates of the continuous and dispersed phases, and the initial concentration of the cell suspension (MCF7 cells, **Figure S1**). To ensure a high fraction of droplets contained a single cell, we designed the droplet-loading chip to order cells using a spiral geometry that achieves inertial focusing of the cells for subsequent ordered loading of droplets (**Figure S2**), as has been reported recently^{39,40}. Single-cell occupancy was achieved in $79.5 \pm 6.1\%$ ($n = 3$) of droplets generated under the following empirically determined conditions: droplet diameter = $\varnothing_{droplet} = 40\text{-}60 \mu\text{m}$; volumetric flow rate of dispersed phase = $Q_{dispersed} = 0.2\text{-}20 \mu\text{L}/\text{min}$; volumetric flow rate of continuous phase = $Q_{continuous} = 0.2\text{-}100 \mu\text{L}/\text{min}$; starting concentration of cell suspension = $1.0\text{-}6.0 \times 10^6 \text{ cells}/\text{mL}$. The highest fraction of single-cell containing droplets was achieved when the starting concentration of the cell suspension was $4.0 \times 10^6 \text{ cells}/\text{mL}$.”

Addition of new works cited:

(39) Moon, H.-S.; Je, K.; Min, J.-W.; Park, D.; Han, K.-Y.; Shin, S.-H.; Park, W.-Y.; Yoo, C. E.; Kim, S.-H. Inertial-ordering-assisted droplet microfluidics for high-throughput single-cell RNA-sequencing. *Lab on a Chip* **2018**, *18* (5), 775-784.

(40) Tang, T.; Zhao, H.; Shen, S.; Yang, L.; Lim, C. T. Enhancing single-cell encapsulation in droplet microfluidics with fine-tunable on-chip sample enrichment. *Microsyst Nanoeng* **2024**, *10*, 3. DOI: 10.1038/s41378-023-00631-y

Page 24 of the **Materials & Methods** section now reads: **“Deterministic droplet-encapsulation of inertially ordered cells.** To achieve a high fraction of droplets containing a single mammalian cell, we adopted a technique that deterministically orders cells (into a line) as the cells are approaching the droplet-encapsulation region of the device. Reported by our previous work⁷⁶, the method employs sigmoidal microchannels with alternative curvatures. In this stage, the channel Reynold number was estimated to be 17 when the cell flow rate was 20 $\mu\text{L}/\text{min}$. The specific channel design and flow parameters in the inertial focusing stage enable the focusing of the cells using inertial lift and Dean drag forces.”

The **Supplemental Information** section now includes a new **Figure S2** which reports the inertial-focusing chip design accompanied by fluorescence micrographs of the deterministic cell-ordering function:

Figure S2. Deterministic droplet-encapsulation of inertially ordered cells. (Top) CAD schematic of droplet-generation chip designed to utilize inertial focusing conditions to order single cells in a ‘single-file queue’ prior to loading cells into droplets at a target occupancy of one cell per droplet. (Bottom) Fluorescence micrographs from chip regions demarcated with green boxes (in top CAD schematic) show inertial focusing of a stream of cells expressing red fluorescence signal (left) and the downstream, ordered flow of cells prior to encapsulation in droplets (right).

Reviewer comment 2: *The scale of the assay is not sufficient to effectively survey cell heterogeneity in a tissue. In contrast, standard cytometry achieves kHz processing, making use of all available cells in the tissue section.*

Authors' response 2: The Reviewer is correct in noting the DropBlot assay was not designed to “effectively survey the cell heterogeneity in a tissue”. Importantly, the DropBlot assay was designed to minimize consumption of archived biospecimens for single-cell protein proteoform analysis, thus leaving substantial material in the biorepository for future analyses. While we highlighted this unmet need and design goal in the **Introduction** of the submitted manuscript, we now clarify the DropBlot design goals through the revisions outlined here:

Page 4 of the **Introduction** now reads: “Microfluidic large-scale integration (mLSI) platforms¹³, single-cell barcode chips (SCBCs)¹⁴, and droplet-based cell screening & sorting¹⁵ reduce the number of cells

required for protein analysis, as compared to flow cytometry and conventional MS. New measurement methods¹⁶ may improve protein detection sensitivity and specificity in fixed cell and tissue samples. When considering analysis of biospecimens archived in biorepositories, limiting consumption of sparingly available sample masses – while maximizing detection sensitivity and specificity – can emerge as a central design tradeoff.

While not optimized for analysis of sparingly available biospecimens, slab-gel immunoblotting is a workhorse targeted-proteomic method with a specificity that is sufficient to detect protein proteoforms and protein complexes. A type of immunoblot, western blotting couples protein polyacrylamide gel electrophoresis (PAGE) with subsequent immunoassays¹⁷. Slab-gel immunoblotting of fixed samples has been reported^{7, 8}, often pooling tissue or cell samples to enhance detection sensitivity. Our lab introduced a suite of immunoblotting tools optimized for individual unfixed (fresh) cells^{9, 10}, including for analysis of human-derived dissociated solid tumor¹¹ and circulating tumor cell specimens¹².”

Page 5 of the **Introduction** now reads: “Consequently, we introduce a hybrid microfluidic tool – called DropBlot – that extends the relevance of open-fluidic chip design to the preparation and analysis of single fixed cancer cells, and aims to conserve the use of precious, archived biorepository specimens.

DropBlot combines droplet-based, single-cell sample preparation with open-fluidic, single-cell western blotting for 100’s of chemically fixed cells, as is directly relevant to archived biospecimens.”

Reviewer comment 3: *The droplet format has a further disadvantage in that washing steps cannot be used - the use of 6M urea produces multiple peaks per protein is one point of concern.*

Authors' response 3: We thank the Reviewer and elaborate on the approach of using urea to aid antigen retrieval from chemically fixed cells via DropBlot. We make the following revisions:

Page 17 of the **Results & Discussion** now reads: For DropBlot analysis of PFA-fixed cells, we explored the impact of increasing the fixation time (15 vs. 30 min) and increasing the antigen-retrieval incubation time (120 min) and observed that increasing the fixation time (30 min) dramatically reduced the number and intensity of detectable protein peaks (Error! Reference source not found.c-e, **Figure S13**). Extended PFA fixation can increase the crosslinking strength between proteins and lipids, making antigen retrieval difficult⁶⁴. Further, we sought to achieve a balance between enhanced antigen-target retrieval with elevated incubation temperatures versus deleterious effects that are exacerbated at elevated temperatures in the presence of urea (i.e., carbamylation, protein aggregation)^{65,66}. To mitigate carbamylation during antigen retrieval in the presence of 6 M urea, we reduced incubation temperatures (below 37°C) and acidified the urea solution by adding 100 mM HCl. Acidification helps impede the formation of isocyanic acid, thereby minimizing carbamylation. However, we were unable to arrive at acceptable antigen retrieval performance for DropBlot protocols that included 6 M urea, thus deferring optimization of urea conditions to future assay development efforts.”

Page 20 of the **Results & Discussion** now reads: “The antigen-retrieval buffer used in our experiments (0.5% (w/v) SDS, 6 M urea), may not complete cell lysis and efficient release of the target proteins. Based on these survey results, enzymatic antigen retrieval is a topic of future study for DropBlot. Incubating cells with urea at elevated temperatures can lead to protein aggregation and carbamylation, leading to adverse impacts on solubility and immunoreactivity of retrieved antigen targets.

Addition of new works cited:

(65) Gorisse, L.; Pietrement, C.; Vuiblet, V.; Schmelzer, C. E.; Köhler, M.; Duca, L.; Debelle, L.; Fornès, P.; Jaisson, S.; Gillery, P. Protein carbamylation is a hallmark of aging. *Proceedings of the National Academy of Sciences* **2016**, *113* (5), 1191-1196.

(66) Sun, S.; Zhou, J.-Y.; Yang, W.; Zhang, H. Inhibition of protein carbamylation in urea solution using ammonium-containing buffers. *Analytical biochemistry* **2014**, *446*, 76-81.

Reviewer comment 4: *Also, the multiplexing capability given available space and multi-peak separations is limited and will be rivalled by cytometry (up to 20-plex) or mass cytometry (up to 40-plex). One merit of single-cell blotting is the ability to identify isoforms with a single antibody). However, antibodies specific to different isoforms (of research and clinical value) are commercially available.*

Authors' response 4: We thank the Reviewer for articulating the DropBlot design tradeoffs in a manner that is more effective than our communication in the original submission. For clarity, DropBlot was *not* designed to profile cellular heterogeneity across a solid tumor. DropBlot is designed to be more similar to FACS analysis – with the added capacity to profile protein isoforms and protein complexes from single cells – than to IHC. To clear up misunderstanding of the design goals of DropBlot, we modify the **Results & Discussion** and **Supplementary Information** as follows:

Page 5 of the **Results & Discussion** section now reads: “While relevant to readily lysed fresh cells, the open-microwell design supports only brief cell-lysis durations (<1 min, <50°C) before the lysate dilutes and diffuses out of the microwell, thus making the readily translatable tools irrelevant to preparation of fixed cells that require long-duration (>60 min) and harsh antigen retrieval conditions^{21, 22}. **Benchmarking of contemporary single-cell proteomic analysis techniques and associated performance tradeoffs are summarized in Table S1.** Consequently, we introduce a hybrid microfluidic tool – called DropBlot – that extends the relevance of open-fluidic chip design to the preparation and analysis of single fixed cancer cells, and aims to conserve the use of precious, archived biorepository specimens. DropBlot is designed to provide target specificity suitable for proteoforms and protein complexes with single-cell resolution. These protein-target classes often lack specific antibody probes. As a corollary design goal, the DropBlot assay is designed to limit consumption of sparingly available archived, fixed biospecimens, hence the use of microfluidic design.”

Page 23 of the **Results & Discussion** section now reads: “Based on these results and demonstrated sufficient performance for PFA- and methanol-fixed cells, we see the potential to further mature and refine DropBlot, including exploring additional sample preparation conditions to determine if additional fixation chemistries are compatible with the sample preparation-to-analysis workflow. **While promising for the dual design goals of (i) analysis of sparingly limited cell specimens from biorepositories and (ii) proteoform detection, several areas for obvious performance enhancement and optimization exist, depending on the application area of interest. For example, an immediate next goal could entail expanding target-detection multiplexing beyond the handful of targets detected in each cell that is reported in this first report. Another area ripe for innovation is in antigen retrieval from formalin-fixed, paraffin-embedded (FFPE) cell specimens.** In terms of system design, while DropBlot is utilized here for serial, complementary unit functions (sample preparation to analysis or analysis to detection), we see the modular design as potentially quite powerful for multi-omics questions where coupling of different, complementary analysis stages performed on the same individual cell may lead to breakthroughs in understanding.”

Page 31 of the Materials & methods section now reads: “**Immunoprobng and fluorescence imaging for the cell analysis step.** The primary antibody immunoprobng solution was prepared by diluting stock solutions of primary antibodies in 2% (w/v) BSA/TBST solution to achieve an antibody concentration of 0.05 µg/µL (single antibody). Primary antibodies used were EpCAM, VIM, HER2, and GAPDH (Abcam, Cambridge, United Kingdom). The single-cell western blotting device (gel slide) was treated with 80 µL of primary antibody immunoprobng solution and incubated at room temperature for 2 hr. After incubation, each single-cell western blotting device was washed twice with TBST buffer for 1 hour. The secondary antibody immunoprobng solution was prepared by diluting stock solutions of primary antibodies in 2%

(w/v) BSA/TBST solution to achieve a concentration of 0.05 $\mu\text{g}/\mu\text{L}$ (single antibody). The single-cell western blotting device was incubated with 80 μL of secondary antibody immunoprobng solution at room temperature for 2 hr. After incubation, the single-cell western blotting device was washed twice with TBST buffer for 1 hr. Before fluorescence imaging, the single-cell western blotting device was washed 3 \times with DI water to remove excess salts, and dried with nitrogen gun. The single-cell western blotting device was imaged with a Genepix Microarray Scanner. Images were analyzed using custom analysis scripts in MATLAB (MathWorks, Natick, MA). Two to three protein targets were analyzed concurrently. Antigen-target multiplexing utilized an established stripping and re-probing method⁸⁰.”

As referenced above, the **Supplementary Information** has been revised to include a landscape analysis of performance tradeoffs of contemporary single-cell protein assays (**Table S1: Current Single-Cell Proteomics Analysis Techniques**).

Table S1: Current Single-Cell Proteomics Analysis Techniques

Name	Method	Fixed/ Live	Target	Surface /Intracellular Protein	Proteoforms detection	Throughput	Multiplexity	Pros	Cons	Ref
Mass Spectrometry (Top down)	Direct analysis of intact proteins	Live (fixed cells need new lysis step)	Protein	Both	Yes	<100 cells	>1000	able to intact protein molecules, high sensitivity.	Low throughput, Limited to small-to-intermediate proteins (<25 kDa); difficulty to distinguish proteoforms due to high sample complexity	24-26
Mass Spectrometry (Bottom up)	Direct analysis of digested proteins	Live (fixed cells need new lysis step)	Protein	Both	Yes	<100 cells	>1000	Highly multiplexed	Low throughput, limited to high abundant proteins (>10,000 copies /cell); Protein digestions will miss proteoform stoichiometry	27-29
cyTOF (Cytometry by time of flight)	Flow cytometry and inductive coupled plasma mass spectrometry; Use metal isotope tagged antibodies.	fixed cells, fixed tissues	Protein	both, intracellular proteins analysis requires cell fixation	Yes, based on the availability of proteoform antibody	~ 100 cells /s	~100	Highly multiplexed (>100 protein targets); Low background	Cannot be applied to live cells; Limited availability of commercial metal-isotope-labelled antibodies; difficult to analyze single cells due to low recovery rate and high sample loss.	30, 31
FCM (flow cytometry)	Label based, rely on fluorescent signals	live or fixed	Protein and nucleic acids	Both	Yes, based on the availability of proteoform antibody	100-10,000 cells /s	~17	high throughput, highly multiplexed	Requires large sample. Low sensitivity due to spectra overlap and autofluorescence. Limited proteoform antibodies	32-34
IMC (Imaging Mass Cytometry)	Labeled with isotope conjugated antibodies, and analyzed with Mass Spectrometry)	Fixed / Frozen Tissue	Protein	Both	Yes	1 mm ² / 2h	~40	High sensitivity, highly multiplexed; advance in spatial resolution	Low throughput; Limited to small proteins (<20 kDa), Limited proteoform antibodies; High Cost;	35, 36
(PiMS) Proteoform Imaging Mass Spectrometry	Combination of nanospray desorption electrospray (nano-DESI) and individual ion MS (I ² MS)	Fixed Tissue	Protein	Both	Yes	2.5-4 um /s	~169 (proteoforms)	highly multiplexed; High spatial resolution	Limited to proteins smaller than 70 kDa	37
Abseq	label-based, antibodies are labeled with sequence tags	Live (compatible to fixed cellss)	Protein	Surface	Not applied yet, but in theory it can.	10000 cells / 1h	unlimited, but can be reduced based on the available proteins, and reading capacity)	high sensitivity to low-abundance antibodies (single molecule per cell);	some antibodies cannot be labeled with detectable tags	38
On-Chip Cytometry (Microengraving)	Label-Based, rely on fluorescent signals.	Live	Protein and nucleic acids	Surface	Yes	84,672 cells/ array	~4	parallel study; Cells can be recovered.	Limited to availability of proteoform antibodies.	39
Single-cell barcode chips (SCBCs)	Label-Based, rely on fluorescent signals.	Live/Fix	Protein	secreted proteins	Not applied yet, but in theory it can.	3000-5000 cells/ array	~42	parallel study; Cells can be recovered. Highly multiplexed	Limited to secreted proteins	14
Quantitative Ferrohydrodynamic Cell Separation (qFCS)	Label-based, cells are labeled with magnetic beads and sorted based on the antigen density	Live/Fix	Protein	Surface	No	30,000 cells/min	1	can detect rare cell types, as low as 10 cells/mL; high sensitivity to low abundance antigens.	Low multiplexed. Cannot analyze proteoforms.	16

single cell western blot (scWB, 2D)	First separate based on molecular weight, and then use fluorescent antibodies to visualize protein targets	Live	Protein	Both	Yes	~5000 cells/array	~12	Parallel study; Capable of analyzing proteoforms.	Low multiplexed.	12, 40
single cell western blot (scWB, 3D)	First separate based on molecular weight, and then use fluorescent antibodies to visualize protein targets	Live	Protein	Both	Yes	2.5 cells /s, 300 cells/array	~4	Parallel study; Low sample consumption	Low multiplexed. A large number of images to process. Cannot be applied to fixed cells	41
Magnetic ranking cytometry (MagRC)	Label-based, cells are labeled with magnetic nanoparticles.	Live/Fix	Protein	Surface	No	500 ul/h	1	suitable to rare cells.	Low multiplexed.	42
Droplet-based cell screening & sorting	Label-based, proteins are labeled with fluorescence antibodies	Live	Protein	Surface	No	2 - 5e5 cells/h	1	high sensitivity, minimal cross-contamination	Low multiplexed.	43, 44
Digital microfluidics (DMF)	Digital microfluidics provide single cell sample (protein, nucleic acids) for downstream analysis (e.g., LC-MS/MS); Sample preparation is in droplet	Live/Fix	Protein/nucleic acids	Both	Not applied yet, but in theory it can.	50-500 cell / assay	>1000	High precision, low sample consumption and ability of perform complex manipulation of small volumes of liquid	Low throughput; complicated; target detection relies on other techniques (e.g., MS).	45-47
oil-air droplet (OAD) chip	Combination of droplet microfluidics and LC-MS/MS	Live	Protein	Both	Not applied yet, but in theory it can.	1-100 cells/chip	~355	Low sample loss, high sample injection efficiency	Low throughput; complicated; target detection relies on other techniques (e.g, MS).	48
Nanodroplet processing in one-pot (nanoPOTS)	Combination of droplet microfluidics and LC-MS/MS	Live	Protein	Both	Not applied yet, but in theory it can.	10-240 cells / assay	670-3000	highly multiplexed; low sample contamination	Low throughput; complicated; target detection relies on other techniques (e.g., MS).	29, 49
Single-cell integrated proteomic microfluidic chip (SciProChip)	Combination of on-chip peptide preparation and LC-MS/MS	Live	Protein	Both	Not applied yet, but in theory it can.	20 / assay	~1500	highly multiplexed; low sample contamination	Low throughput; complicated; target detection relies on other techniques (e.g., MS).	50
Immunohistochemistry (IHC) or Immunocytochemistry (ICC)	Label-based, proteins are labeled with antibodies and visualized with colored chromogen or fluorophores	Fixed	Proteins	Both	Yes	NA	~2-50	High specificity, tissue localization, wide applicability,	Limited by the availability of antibodies, narrow dynamic range, time consuming, variability	51-53
DropBlot	Combination of droplet microfluidics with single-cell western blotting	Live/Fix	Protein	Both	Yes	~5000 cells/assay	~20	Robust; capable of analyze proteoforms in fresh/fixed cells; minimized sample loss	Lysis buffer is limited due to the droplet stability. Low multiplicity; complicated	This work

Reviewer comment 5: *In summary, the manuscript documents good progress in tackling the different challenges of processing fixed cells in droplets and interfacing this with a single-cell electrophoresis array. Despite the good progress, the method and overall concept is unlikely to replace existing approaches and make an impact in the field which places increasing demands on throughput and multiplexing.*

Authors' response 5: Thank you. These helpful comments made us realize that our initial manuscript submission did not clearly communicate the design goals of the DropBlot (similar to our response above): which are not to advance multiplexing or throughput. We apologize. Since the concern is similar to that conveyed in Reviewer comment 4, we reiterate that DropBlot aims to perform more like a FACS targeted analysis – with the added capacity to profile specific protein proteoform targets from single cells – than a highly multiplexed discovery proteomics tool. To clear up misunderstanding of the design goals of DropBlot, we modify the **Results & Discussion** and **Supplementary Information** as follows:

Page 5 of the Results & Discussion section now reads: “Consequently, we introduce a hybrid microfluidic tool – called DropBlot – that extends the relevance of open-fluidic chip design to the preparation and analysis of single fixed cancer cells, and aims to conserve the use of precious, archived biorepository specimens. DropBlot is designed to provide target specificity suitable for proteoforms and protein complexes with single-cell resolution. These protein-target classes often lack specific antibody probes. As a corollary design goal, the DropBlot assay is designed to limit consumption of sparingly available archived, fixed biospecimens, hence the use of microfluidic design.”

Page 18 of the **Results & Discussion** section now reads: “Based on these results and demonstrated sufficient performance for PFA- and methanol-fixed cells, we see the potential to further mature and refine DropBlot, including exploring additional sample preparation conditions to determine if additional fixation chemistries are compatible with the sample preparation-to-analysis workflow. While promising for the dual design goals of (i) analysis of sparingly limited cell specimens from biorepositories and (ii) proteoform detection, several areas for obvious performance enhancement and optimization exist, depending on the application area of interest. For example, an immediate next goal could entail expanding target-detection multiplexing beyond the handful of targets detected in each cell that is reported in this first report. Another area ripe for innovation is in antigen retrieval from formalin-fixed, paraffin-embedded (FFPE) cell specimens. In terms of system design, while DropBlot is utilized here for serial, complementary unit functions (sample preparation to analysis or analysis to detection), we see the modular design as potentially quite powerful for multi-omics questions where coupling of different, complementary analysis stages performed on the same individual cell may lead to breakthroughs in understanding.”

Reviewer comment 6: *Please note occasional typos were spotted.*

Authors' response 6: We thank the Reviewer for noting minor typographical errors and have pored over the manuscript, making the following corrections.

- Page 9: changed "engineering" to "engineered"
- Page 13: changed “to microwell” to “to the microwell”
- Page 19: change “no signal” to “undetectable signal”
- Page 19: change “any enzymatic antigen-retrieval method” to “all enzymatic antigen-retrieval methods”
- Page 24: change “are presented” to “is presented”
- Page 26: change "50 um droplets" to "50-um diameter droplets"
- Page 27: change “HAMAMATSU PHOTONICS” to “Hamamatsu Photonics”
- Page 28: change “300V constant” to “a 300-V constant”
- Page 28: change “45s’ UV exposure” to a 45-s UV exposure”
- Page 28: change “300V constant” to “a 300-V constant”
- Page 29: change “for three times and aliquoted to a 1.5 mL” to “three times and aliquoted into a 1.5 mL”

- Page 29: change “in 45 um droplet” to “in 45- μ m diameter droplets”
- Page 30: change “in water bath” to “in a water bath”

Response to Reviewer #3:

Reviewer preface comment: *The authors have developed a method to perform single-cell western blots. I will specifically to [sic] comment on the application of this method to breast cancer.*

Authors' preface response: We deeply appreciate input from a breast-cancer expert, even at this early stage. We are focused on communicating contributions in this very first report of DropBlot that are firmly centered in chemistry and engineering (versus cancer biology). The journal's scope statement names both chemistry and engineering as scope-fulfilling disciplines.

Reviewer comment 1: *In figures 5&6, the authors demonstrate the expression of EPCAM, Vimentin and Her2 in unfixed and fixed cells, respectively. These are relevant markers for breast cancer cell lines as they are biologically relevant and often heterogeneously expressed. These figures appear to technically validate the method but do not reveal new biological insights.*

Authors' response 1: We agree with the Reviewer. Figures 5 and 6 are, by design, documenting validation. Figures 5 and 6 are not designed to provide biological insights. As this Reviewer aptly points out in comment 2 (next comment): the “*authors apply this same panel of antibodies to cryopreserved clinical samples*” in Figure 7 as is used in Figures 5 & 6. This methodical progression is, in fact, the assay validation process itself. To clearly communicate the cornerstone that the Figure 5 and 6 validation studies are to introduction of this new life-sciences tool, we make the following revisions:

Page 6 of the **Results & Discussion** now reads: “We develop the DropBlot workflow using two breast cancer cell lines and a suite of chemical fixation conditions. Each protein target is first individually identified by immunoreactivity from fresh-cell (unfixed) lysate and then under each fixation condition for a well-characterized pair of cancer cell lines. We expect each protein target to vary in electromigration behavior depending on the fresh or fixed state of the originating cell, thus methodical assay development and target-identity tracking are utilized. After validating the identity and immunoreactivity of each protein target, we next apply DropBlot to investigate a pilot cancer-protein panel composed of an epithelial marker (EpCAM), a mesenchymal marker (vimentin, VIM), and human epidermal growth receptor 2 (HER2) in a pilot group of breast cancer patient-derived cell samples.”

Page 14 of the **Results & Discussion** now reads: “To assess the relationship between antigen-retrieval buffer formulation (including SDS) and the immunoreactivity of any retrieved antigen, we scrutinized fresh cells from well-studied cultured cell lines and the protein targets epithelial cellular adhesion molecule (EpCAM, 35-40 kDa) and intermediate filament protein (VIM, 57 kDa). We anticipate the chemical state of each originating cell will impact the degree of antigen retrieval for each protein target. Antigen retrieval means, most obviously, the amount of antigen material recovered from a chemically fixed cell. Importantly, and more subtly, antigen retrieval also depends on the degree of recovery of immunoreactivity, which in turn depends on the physicochemical properties of each retrieved antigen species. In this latter aspect, we anticipate that fixation-induced alterations in protein physicochemical properties will inherently affect electromigration. Consequently, we start by establishing protein-target identity in the simplest to the most complex matrix conditions using a modified ‘spiked recovery’ immunoassay development process to account – as much as possible – for matrix effects anticipated in human-derived and chemically fixed cell specimens³. Here, that means we start with target antigen measurement from a clear buffer, progressing next to antigen measurement from fresh cell lines, then retrieval from fixed cells from the same cell lines, and finally move to considering retrieval of a panel of protein targets in fixed patient-derived tumor specimens.”

Reviewer comment 2: *In Figure 7, the authors apply this same panel of antibodies to cryopreserved clinical samples. While they observe heterogenous expression, this is well documented previously using methods such as immunofluorescence. Two of these markers, Epcam and Vim, are not of any clinical value. There are many methods to make single cell protein measurements in clinical samples, using methods based on multiplexed immunofluorescence, that can scale up to 100 antigens. It seems that the specific advantage of this method would be in identifying different molecular weight forms of proteins, but the authors have not chosen targets where that information is relevant or important and have not followed up on the relevance of the 2 Vimentin forms reported.*

Authors' response 2: We appreciate the Reviewer's comments that "the specific advantage of this method would be in identifying molecular weight forms of proteins". While we were at first disappointed in the Reviewer's impression, after considering the Reviewer's excellent point and a similar sentiment from the Editor, we realized that DropBlot presents a powerful functionality for deep-proteoform profiling of selected cell sub-populations that was not communicated in the original submission. Consequently, in response to Reviewer #3, we performed additional analyses, and now report on "FACS-like" gating and selection of HER2+ and then EpCAM+ cell subpopulations with VIM proteoform profiling in each of these two selected cell sub-populations. The new analysis reveals both a new gating analytical capability of DropBlot and a new biomedical insight regarding Sample #3 and the existence of a rare cell sub-population that would not be detectable with existing tools.

As such, we revise the **Abstract**, the **Results & Discussion** by significantly revising **Figure 7 (new data and analysis)** and revise the **Supplementary Information** as follows:

Page 2 of the **Abstract** section now reads: "To further realize proteomics of archived tissues for translational research, we introduce a hybrid microfluidic platform for high-specificity, high-sensitivity protein detection from individual chemically fixed cells. To streamline processing-to-analysis workflows and minimize signal loss, DropBlot serially integrates sample preparation using droplet-based antigen retrieval from single fixed cells with unified analysis-on-a-chip comprising microwell-based antigen extraction followed by chip-based single-cell western blotting. A water-in-oil droplet formulation proves robust to the harsh chemical (SDS, 6 M urea) and thermal conditions (98°C, 1-2 hr.) required for sufficient antigen retrieval, and the electromechanical conditions required for electrotransfer of retrieved antigen from microwell-encapsulated droplets to single-cell electrophoresis. Protein-target retrieval was demonstrated for unfixed, paraformaldehyde- (PFA), and methanol-fixed cells. We observed higher protein electrophoresis separation resolution from PFA-fixed cells with sufficient immunoreactivity confirmed for key targets (HER2, GAPDH, EpCAM, Vimentin) from both fixation chemistries. Multiple forms of EpCAM and Vimentin were detected, a hallmark strength of western-blot analysis. DropBlot of PFA-fixed human-derived breast tumor specimens (n = 5) showed antigen retrieval from cells archived frozen for 6 yrs. In this pilot study, the proteoform-resolving capacity of DropBlot revealed a rare subpopulation of VIM proteoform co-expression in HER2+ cells derived from an invasive ductal carcinoma tissue specimen. DropBlot could provide a precision integrated workflow for single-cell resolution protein-biomarker mining of precious biospecimen repositories."

Pages 20-23 of the **Results & Discussion** section now reads: "**DropBlot analysis of fixed clinical specimens reports rare co-expression of two VIM proteoforms in a sub-population of HER2+ tumor cells.** As a first demonstration of the DropBlot assay after assay development and validation, we focused on analysis of a well-understood protein panel in chemically fixed human-derived biospecimens. As mentioned, the panel consisted of an epithelial marker (EpCAM), a mesenchymal marker exhibiting proteoforms (VIM', VIM''), and human epidermal growth receptor 2 (HER2). In particular, HER2 and VIM were chosen because tumor cells with a mesenchymal phenotype and high expression of HER2 tend to be more aggressive^{4, 5}.

We applied Dropblot to scrutinize single, fixed cells dissociated from 11 solid breast tumor specimens. These human-derived tumor tissues were archived for >6 yrs. stored under -80°C conditions without chemical fixation. Prior to DropBlot analysis, these patient-derived cells were thawed, tissue was dissociated, and PFA-fixation completed (**Figure 7a, Table S5**). DropBlot successfully retrieved antigen from 5 of the PFA-fixed cell specimens, as determined by probing for markers of epithelial-to-

mesenchymal transition (EMT) and tumor cell growth at the single-cell level (**Figure 7b-c**). Having now validated DropBlot on complex human-derived tissue specimens, we sought to understand the sources of VIM-proteoform expression heterogeneity in these specimens, at the level of originating cell type with a particular interest in HER2+ cancer cells (**Figure 7d**).

To distinguish cancer cells from among the other cell types present (i.e., stromal, immune), we applied DropBlot in a 'cell gating' mode – analogous to the gating functionality commonly used in flow cytometry. Gating on protein marker expression allows analysis of specific cellular sub-populations in a manner like flow cytometry, with DropBlot enabling analysis of protein proteoforms at the single-cell level, even when antibody probes specific to each proteoform have either poor performance or are nonexistent. This latter functionality is not possible using flow cytometry, or other existing single-cell immunoassays (i.e., immunofluorescence, IHC, mass cytometry).

Employing DropBlot in cell gating mode to hone in on HER2+ cancer cells, we were particularly interested in cancer cells as sources of VIM and VIM-proteoform expression and heterogeneity (**Figure 7e-g**). We identified HER2+ cell sub-populations across the five human-derived tumor specimens from which antigen was successfully retrieved after chemical fixation. Attending to Sample #3 in a moment, Samples 1 and 2 were fresh cell suspensions and contained 26.4% and 38.9% HER2+ cells, respectively. While 26.4% of Sample #1 cells and 31.4% of Sample #2 cells co-expressed HER2 and VIM', there were no cells detectable that expressed the VIM'' proteoform (either alone or with VIM). Samples 4 and 5, which were suspensions of fresh dissociated tissues, showed 46.3% and 0% cells as HER2+, respectively. In Sample #4, 40.8% cells co-expressed HER2+ and VIM+, with VIM expression arising only from the VIM' proteoform. VIM'' was not detected in Sample #4, either alone or co-expressed with VIM'. Sample #5 reported a cell sub-population of 41.9% of the cells analyzed, which was expressing VIM' alone and likely stromal in origin.

Returning to Sample #3, the 707 cells analyzed were composed of 8.5% HER2+ cells by DropBlot. DropBlot detected HER2+ with no co-expression of VIM in ~5.4% of the cells assayed. Nearly ~2.7% of HER2+ cells expressed one but not the other VIM proteoform (1.3% VIM' only and 1.4% VIM'' only). Importantly, DropBlot detected a rare cell sub-population of ~0.4% of the analyzed cells that were HER2+ and co-expressed both VIM' and VIM'' proteoforms.

Importantly, Sample #3 was classified as an invasive ductal carcinoma. Among the HER2+ cell sub-populations surveyed in this pilot study, Sample #3 proved to be the most heterogeneous in VIM proteoform expression. Previous research shows that the mesenchymal phenotype and high expression of HER2 tend to be indicative of a more aggressive phenotype^{4, 5}. Further, previous studies suggest that the most prevalent form of VIM is the 55-kDa intermediate filament³, with truncated VIM generated during cancer metastasis⁵⁴⁻⁵⁶.

Applying DropBlot and gating on sub-populations of non-cancerous cells, we hypothesized that cells expressing VIM only (either form; but no HER2) may represent stromal cells. Gating on total VIM expression, DropBlot reported ~50% of cells analyzed from Sample #3 expressed VIM with co-expression of no other panel marker for this tumor dissociate. Given the prevalence of the VIM-only expressing cells, stromal cells are the likely originating cell type, as would be expected in a dissociated tumor. In the VIM+ stromal cells of Sample #3, 30.8% expressed VIM', 8.5% expressed VIM'', and 10.3% co-expressed VIM' and VIM'' (**Figure S16**).

In a novel function of the DropBlot assay, we next gated the cell population on expression of just one of the VIM proteoforms. Just ~1.3% of the cells analyzed from Sample #3 co-expressed VIM'' with HER2 and ~9.6% co-expressed VIM' with EpCAM. Next gating on expression of the second VIM'' proteoform only, DropBlot identified ~1.4% of the cells analyzed from Sample #3 as co-expressing VIM'' with HER2 and ~5.2% co-expressing VIM'' with EpCAM. Roughly 14.9% of the Sample #3 cells analyzed were positive for VIM (VIM' or VIM'') and EpCAM, with ~2.3% expressing both VIM proteoforms and EpCAM.

Figure 7. DropBlot analysis of single PFA- and methanol-fixed patient-derived dissociated cancer cells. (a) Schematic of clinical sample preparation workflow for DropBlot. PFA conditions: $\Delta t_{\text{fixation}} = 15 \text{ min}$; $\Delta t_{\text{incubation}} = 1.0 \text{ h}$ at 98°C ; Figures generated with BioRender. (b) Fluorescence micrographs of single-cell western blots of EpCAM (Green, AF488-labeled secondary antibody), VIM (Red, AF594-labeled secondary antibody), and HER2 (Blue, AF647-labeled secondary antibody) from PFA-fixed tumor cell. Tumor was classified as triple-positive breast cancer. (c) Mean fluorescence intensity of single-cell western blot analyses of PFA-fixed, patient-derived tumor cells for EpCAM, VIM proteoforms (VIM', VIM''), and HER2. Samples #1-2 were fresh cell suspensions. Sample #3-5 were fresh dissociated tissues. The tumor cells were identified as EpCAM+ or HER2+. (d) Fluorescence micrographs of single-cell western blots of PFA-fixed cells from Sample #3 from (c), with HER2 (blue, AF647-labeled secondary antibody) and VIM (red, AF594-labeled secondary antibody). (e) Cell gating using DropBlot. HER2+ positive cells were further classified based on the expression levels of VIM' and VIM''. The protein target was considered as negative when the intensity was less than 4. (f) Cell gating using DropBlot. EpCAM+ positive cells were further classified based on the expression levels of VIM' and VIM''. The protein target was considered as negative when the intensity was less than 4. (g) Venn diagram reporting the single-cell target-expression profile for each of single PFA-fixed cells from Sample #1-5 in (c).

The **Supplemental Information** section now includes a new **Figure S16** to introduce the VIM expressions in EpCAM-/HER2- cells.

EpCAM-/HER2- Cells

Total: 351

Figure S16. Venn diagram reports the single-cell target-expression profile for PFA-fixed EpCAM-/HER2- cells ($n = 351$) from Sample #3 in Figure 7c.

Reviewer comment 3: *I am concerned that the authors have taken whole tumour dissociates, which will contain not just cancer cells but also stromal cells (which are mostly vimentin+) and immune cells (which are mostly negative for all markers). Therefore the reported observation of heterogenous expression of proteins is likely to be a product of varying cellular composition, rather than heterogenous expression by cancer cells as the authors conclude. For instance, the 2 vimentin isoforms may be expressed by epithelial vs stromal cells. The authors do not validate their findings.*

Authors' response 3: We appreciate the Reviewer's critical and insightful suggestion, and have notably revised data and analyses presented in **Figure 7** to highlight an important – yet previously undescribed – functionality of the DropBlot that addresses this Reviewer's concern. We revise the **Results & Discussion** (new data & analysis in **Figure 7**) and the **Supplementary Information** to describe DropBlot's ability to distinguish the cell types from cell mixtures (like FACS) and support a deep profile of proteoforms in sorted cells. As addressed in the revisions in response to comments 2 & 3, the assay validation process is documented in detail in the progression through Figures 5 and 6, to Figure 7.

Page 16 of the Results & Discussion section now read: “ We further observed three resolved EpCAM peaks in the lysate of single MCF7 cells, while MDA-MB-231 cells exhibited one detectable peak (**Figure 5e-h**). We attribute the cell-line-dependent EpCAM expression to different proteoforms of EpCAM (**Table S2**). In the intricate interplay of cellular distinctions, DropBlot emerges as a discerning tool, capable of delineating between cell types based on their unique fingerprint proteins and proteoforms (**Figure S11**). Notably, the H1299 lung cancer cell line is demarcated from PBMCs by the distinct presence of CD45 and VIM⁵⁷, while discriminating between MDA-MB-231 cells and stromal fibroblasts can rely on nuanced variations in VIM and fibroblast activation protein (FAP) expression levels and proteoforms. Although FAP is also expressed on MDA-MB-231, the expression level and types of proteoforms vary⁵⁸. The detailed experimental setup is included in the supplementary information (**SI Methods**).”

Pages 20-23 of the **Results & Discussion** section now reads: “ **DropBlot analysis of fixed clinical specimens reports rare co-expression of two VIM proteoforms in a sub-population of HER2+ tumor cells.** As a first demonstration of the DropBlot assay after assay development and validation, we focused on analysis of a well-understood protein panel in chemically fixed human-derived biospecimens. As mentioned, the panel consisted of an epithelial marker (EpCAM), a mesenchymal marker exhibiting proteoforms (VIM', VIM''), and human epidermal growth receptor 2 (HER2). In particular, HER2 and VIM were chosen because tumor cells with a mesenchymal phenotype and high expression of HER2 tend to be more aggressive^{4,5}.

We applied DropBlot to scrutinize single, fixed cells dissociated from 11 solid breast tumor specimens. These human-derived tumor tissues were archived for >6 yrs. stored under -80°C conditions without chemical fixation. Prior to DropBlot analysis, these patient-derived cells were thawed, tissue was dissociated, and PFA-fixation completed (**Figure 7a, Table S5**). DropBlot successfully retrieved antigen from 5 of the PFA-fixed cell specimens, as determined by probing for markers of epithelial-to-mesenchymal transition (EMT) and tumor cell growth at the single-cell level (**Figure 7b-c**). Having now validated DropBlot on complex human-derived tissue specimens, we sought to understand the sources of VIM-proteoform expression heterogeneity in these specimens, at the level of originating cell type with a particular interest in HER2+ cancer cells (**Figure 7d**).

To distinguish cancer cells from among the other cell types present (i.e., stromal, immune), we applied DropBlot in a ‘cell gating’ mode – analogous to the gating functionality commonly used in flow cytometry. Gating on protein marker expression allows analysis of specific cellular sub-populations in a manner like flow cytometry, with DropBlot enabling analysis of protein proteoforms at the single-cell level, even when antibody probes specific to each proteoform have either poor performance or are nonexistent. This latter functionality is not possible using flow cytometry, or other existing single-cell immunoassays (i.e., immunofluorescence, IHC, mass cytometry).

Employing DropBlot in cell gating mode to hone in on HER2+ cancer cells, we were particularly interested in cancer cells as sources of VIM and VIM-proteoform expression and heterogeneity (**Figure 7e-g**). We identified HER2+ cell sub-populations across the five human-derived tumor specimens from which antigen was successfully retrieved after chemical fixation. Attending to Sample #3 in a moment, Samples #1 and #2 were fresh cell suspensions and contained 26.4% and 38.9% HER2+ cells, respectively. While 26.4% of Sample #1 cells and 31.4% of Sample #2 cells co-expressed HER2 and VIM’, there were no cells detectable that expressed the VIM’’ proteoform (either alone or with VIM). Samples 4 and 5, which were suspensions of fresh dissociated tissues, showed 46.3% and 0% cells as HER2+, respectively. In Sample #4, 40.8% cells co-expressed HER2+ and VIM+, with VIM expression arising only from the VIM’ proteoform. VIM’’ was not detected in Sample #4, either alone or co-expressed with VIM’. Sample #5 reported a cell sub-population of 41.9% of the cells analyzed, which was expressing VIM’ alone and likely stromal in origin.

Returning to Sample #3, the 707 cells analyzed were composed of 8.5% HER2+ cells by DropBlot. DropBlot detected HER2+ with no co-expression of VIM in ~5.4% of the cells assayed. Nearly ~2.7% of HER2+ cells expressed one but not the other VIM proteoform (1.3% VIM’ only and 1.4% VIM’’ only). Importantly, DropBlot detected a rare cell sub-population of ~0.4% of the analyzed cells that were HER2+ and co-expressed both VIM’ and VIM’’ proteoforms.

Importantly, Sample #3 was classified as an invasive ductal carcinoma. Among the HER2+ cell sub-populations surveyed in this pilot study, Sample #3 proved to be the most heterogeneous in VIM proteoform expression. Previous research shows that the mesenchymal phenotype and high expression of HER2 tend to be indicative of a more aggressive phenotype^{4,5}. Further, previous studies suggest that the most prevalent form of VIM is the 55-kDa intermediate filament³, with truncated VIM generated during cancer metastasis⁵⁴⁻⁵⁶.

Applying DropBlot and gating on sub-populations of non-cancerous cells, we hypothesized that cells expressing VIM only (either form; but no HER2) may represent stromal cells. Gating on total VIM expression, DropBlot reported ~50% of cells analyzed from Sample #3 expressed VIM with co-expression of no other panel marker for this tumor dissociate. Given the prevalence of the VIM-only expressing cells, stromal cells are the likely originating cell type, as would be expected in a dissociated tumor. In the VIM+ stromal cells of Sample #3, 30.8% expressed VIM’, 8.5% expressed VIM’’, and 10.3% co-expressed VIM’ and VIM’’ (**Figure S17**).

In a novel function of the DropBlot assay, we next gated the cell population on expression of just one of the VIM proteoforms. Just ~1.3% of the cells analyzed from Sample #3 co-expressed VIM’ with HER2 and ~9.6% co-expressed VIM’ with EpCAM. Next gating on expression of the second VIM’’ proteoform only, DropBlot identified ~1.4% of the cells analyzed from Sample #3 as co-expressing VIM’’ with HER2 and ~5.2% co-expressing VIM’’ with EpCAM. Roughly 14.9% of the Sample #3 cells analyzed were

positive for VIM (VIM' or VIM'') and EpCAM, with ~2.3% expressing both VIM proteoforms and EpCAM.

The **Supplemental Information** section now includes a new **Figure S11 and Method** section to introduce the DropBlot's ability to distinguish cell mixtures.

Figure S11. Identification of cell type based protein and proteoform markers detected by single-cell western blot (tPAGE = 30 s; E = 60 V/cm). (a) Single-cell western blot micrographs report differential identification of peripheral blood mononuclear cells (PBMC) and H1299 (fresh) cells using CD45 and Vimentin markers, respectively. (b) Single-cell western blot micrographs report differential identification of MDA-MB-231 and fibroblast (fresh) cells using Vimentin and Fibroblast activation protein- α (FAP) markers, respectively.

REVIEWER COMMENTS

Reviewer #1 (Remarks to the Author):

I was excited by the original manuscript as I felt that this was a significant advance in single protein analysis. I originally had requested some clarifications which have been adequately addressed. This is a fine paper and will stimulate much interest in a hot topic.

Reviewer #2 (Remarks to the Author):

Thank you for responding in depth to my comments.

Point 1. The use of entrainment principles makes sense for surpassing the limitations of random loading, although I'm surprised this key part of the methodology was overlooked in the original submission. Some details require clarification: It is described as a spiral geometry, yet Figure S2 is a serpentine device.

The 79.5% single cell occupancy claimed for MCF7 is incredibly impressive - it is important to understand occupancy numbers when applied to much more challenging fixed tissues which are notoriously difficult to reduce to single cell suspensions.

The new Figure 3S has an image showing cell loading (3×10^6) in which single cell loading numbers are not so impressive.

Can the authors please explain the disparity between the numbers: 50 μm droplets (65 μL) and an optimal cell concentration of 4×10^6 would lead to 1 in 4 droplets containing single cells (at absolute best). This is somewhat distant from the 4 in 5 droplets reported above. How can this difference be reconciled?

The manuscript later reports that 100's of chemically fixed cells to be analysed. Are such numbers (not 79.5% of 5000) more realistic?

References are now given for spiral devices as well as the serpentine device. Some prominent literature is absent and should be considered (Kemna 2012; doi.org/10.1039/C2LC00013J; Harrington 2021; doi.org/10.1039/D1LC00292A; Edd 2008, [10.1039/b805456h](https://doi.org/10.1039/b805456h)).

Point 2. The technology is now described as allowing sparing use of precious samples. Please can the authors expand on sample processing. How do you preserve the remainder of the tissue/what losses occur in tubing, syringes, excess droplets. DropBlot only needs 5000 droplets, or approx. 1 microlitre at 4×10^6 cells, yet handling such small samples for microfluidic applications has not, as far as I am aware, been reported. The manuscript later reports that 100's of chemically fixed cells to be analysed.

Point 3. Thank you for clarifying that urea methods are the topic of future assay developments.

Reviewer #3 (Remarks to the Author):

The authors have done significant work to revise the manuscript to address queries, and the addition of cell gating in clinical specimens is a valuable demonstration of the potential of this technology.

Some of the newly added text is excessively verbose, very detailed and could be edited for brevity and clarity

I have several comments regarding Figure 7:

- I am concerned that some conclusions are based on very small numbers. For instance, the rare subpopulation of cells ("DropBlot detected a rare cell sub-population of ~0.4%...") equates to 3 cells, if I understand correctly. This is simply too few to make a conclusion on, and I recommend removing reference to this in the results and abstract.

- Please indicate the cell numbers that support each conclusion discussed.

- Please show histograms for Her2 and EPCAM alone, with a line showing the threshold used for gating. Why do the plots in 7e & 7f look as though they have been thresholded? I.e, there is not a continuous distribution on either axis. Please show unfiltered histograms for VIM isoforms, both with and without pre-gating for ERBB2/EPCAM

- Regarding this statement "Sample #5 reported a cell sub-population of 41.9% of the cells analyzed, which was expressing VIM' alone and likely stromal in origin": Please explain this further. Does this mean 41.9% of cells were Her2+Vim+? If so, stromal cells are typically not Her2+. Please explain

- Regarding this statment: "Returning to Sample #3, the 707 cells analyzed were composed of 8.5% HER2+ cells by DropBlot. DropBlot detected HER2+ with no co-expression of VIM in ~5.4% of the cells assayed": In the figure it appears to be 63.3% rather than 5.4%. Please clarify

- The paragraph commencing "In a novel function of the DropBlot assay, we next gated the cell population on expression of just one of. the VIM proteoforms." is confusing and does not appear to have a focus. I suggest either deleting it or clarifying, including cell numbers in each population discussed

- A general comment is that the single cell results would be more convincing if bulk immunoblotting data was provided to refer to for each sample analysed

Response to Reviewer #1 (Remarks to the Author):

Reviewer comment 1: *I was excited by the original manuscript as I felt that this was a significant advance in single protein analysis. I originally had requested some clarifications which have been adequately addressed. This is a fine paper and will stimulate much interest in a hot topic.*

Author's Response: Thank you – we are excited about this new technique, as well. Further, we believe that the manuscript is stronger in revision owing to your questions and comments.

Response to Reviewer 2

Thank you for responding in depth to my comments.

Reviewer comment 1: *Point 1. The use of entrainment principles makes sense for surpassing the limitations of random loading, although I'm surprised this key part of the methodology was overlooked in the original submission. Some details require clarification: It is described as a spiral geometry, yet Figure S2 is a serpentine device.*

Author's response 1: We apologize for our oversight and are, frankly, embarrassed to have missed that key communication in the original submission. Consequently, we are grateful that the Reviewer carefully noted the oversight and that we could remedy the accidental omission during peer review. To the suggestion: we agree that the device in **Figure S2** is best described as a *serpentine* geometry and we revise to reflect that more accurate description:

Page 8 of **Result & Discussion** now reads: “To ensure a high fraction of droplets contained a single cell, we designed the droplet-loading chip to order cells using a **serpentine geometry** that achieves inertial focusing of the cells for subsequent ordered loading of droplets (**Figure S2**), as has been reported recently.”

Reviewer comment 2: *The 79.5% single cell occupancy claimed for MCF7 is incredibly impressive - it is important to understand occupancy numbers when applied to much more challenging fixed tissues which are notoriously difficult to reduce to single cell suspensions. The new Figure 3S has an image showing cell loading (3×10^6) in which single cell loading numbers are not so impressive. Can the authors please explain the disparity between the numbers: 50 μm droplets (65 pL) and an optimal cell concentration of 4×10^6 would lead to 1 in 4 droplets containing single cells (at absolute best). This is somewhat distant from the 4 in 5 droplets reported above. How can this difference be reconciled?*

Author's response 2: We thank the Reviewer for requesting clarification. A revised **Figure S3** communicates the role of gravity-based sedimentation in the syringe barrel during the cell-loading process. During loading, the local density of the cell suspension increases at the bottom of the syringe barrel (near the syringe outlet to the chip) due to gravity-based sedimentation of the negatively buoyant cells. We make the following revisions to the **Results & Discussion** and **Material& Method** sections and add a new **Supplemental Figure**, as follows:

Page 8 of **Result & Discussion** now reads: “...starting concentration of cell suspension = $1.0\text{-}6.0 \times 10^6$ cells/mL; **loading time: <15 min.**”

Page 24 of **Materials & Method** now reads: “...In this stage, the channel Reynolds number was estimated to be 17 with a flow rate of 20 $\mu\text{L}/\text{min}$. The channel design and flow parameters focus cells using inertial lift and Dean drag forces. **The cell encapsulation efficiency is affected by gravity-induced changes in the loading density of the cell suspension and will vary over time (Figure S3).** We observed the highest

fraction of single-cell containing droplets with a cell-suspension starting concentration of 4.0×10^6 cells/mL when the loading time is less than 15 min. Noting that the syringe barrel was positioned parallel to gravity, during the minutes-long loading period we observed gravity-induced cell sedimentation, which increases the local concentration of the cell suspension near the syringe outlet, thus affecting the single-cell occupancy rate of the droplets produced⁸⁵. We measured the single-cell droplet occupancy after a 50-min collection period at $12.2 \pm 4.8\%$ ($n = 3$). Thus, to achieve the highest fraction of single-cell occupancy droplets, we suggest a loading duration of <15 min (considering a 10-min setup period). To reduce or even eliminate the observed gravity-induced cell sedimentation, the use of OptiPrep in the medium will reduce the density difference between the cells and the medium⁸⁶.”

Addition of works cited:

[84] Launier, C. A., Czaplowski, G. J., Myung, J. H., Hong, S., & Eddington, D. T. (2011). Rheologically biomimetic cell suspensions for decreased cell settling in microfluidic devices. *Biomedical microdevices*, 13, 549-557.

[85] Mazutis, L., Gilbert, J., Ung, W. L., Weitz, D. A., Griffiths, A. D., & Heyman, J. A. (2013). Single-cell analysis and sorting using droplet-based microfluidics. *Nature protocols*, 8(5), 870-891.

The **Supplemental Information** includes a revised **Figure S3**, which reports the effect of gravity-based sedimentation on the time-variant fraction of droplets that are produced with single-cell occupancy.

Figure S3. Cell encapsulation under the effect of gravity. (a) Schematic illustration of droplet generation and cell encapsulation workflow. Gravity-induced cell sedimentation during the loading period results in a time-variant density of the cell suspension near the syringe outlet. Figure created with BioRender. (b) Cell encapsulation efficiency was measured during the loading period. Cells: fresh MCF7; $Q_{dispersed} = 10 \mu\text{L}/\text{min}$, $Q_{continuous} = 15 \mu\text{L}/\text{min}$. Loading periods 10, 30, 50 min. Cell concentration: 4.0×10^6 cells/mL (c) Bright-field micrographs of resultant droplet populations generated under either 10-min or 50-min cell-suspension loading periods.

Reviewer comment 3: The manuscript later reports that 100's of chemically fixed cells to be analysed. Are such numbers (not 79.5% of 5000) more realistic?

Author's response 3: Thank you for pointing out this unclear statement. We revise the description to limit our discussion to the throughput demonstrated in this manuscript (versus the throughput possible in other embodiments that are not reported here). We revise the **Introduction** and **Materials & Methods** sections and revise **Table S1** in the **Supplementary information** as such:

Page 6 of the **Introduction** now reads: “DropBlot combines droplet-based, single-cell sample preparation with open-fluidic, single-cell western blotting of **100's-1000's** of chemically fixed cells, as is directly relevant to archived biospecimens.”

Page 25 of the **Materials & Methods** now reads: “...Each microwell in the array of 2,500 microwells had a diameter of 50 μm and a depth of 60 μm .”

Revised **Table S1** in the **Supplemental Information** reports the throughput of DropBlot:

DropBlot	Combination of droplet microfluidics with single-cell western blotting	Live/Fix	Protein	Both	Yes	~2,500 cells/assay	~20
-----------------	--	----------	---------	------	-----	--------------------	-----

Reviewer comment 4: *References are now given for spiral devices as well as the serpentine device. Some prominent literature is absent and should be considered (Kemna 2012; doi.org/10.1039/C2LC00013J; Harrington 2021; doi.org/10.1039/D1LC00292A; Edd 2008, 10.1039/b805456h).*

Author’s response 4: We thank the Reviewer and revise the **Results and Discussion** section accordingly to provide a more comprehensive reference list, including those excellent papers suggested by this Reviewer:

Page 8 of **Result & Discussion** now reads: “To ensure a high fraction of droplets contained a single cell, we designed the droplet-loading chip to order cells using a serpentine geometry that achieves inertial focusing of the cells for subsequent ordered loading of droplets (**Figure S2**), as has been reported recently⁴³⁻⁴⁷.”

Addition of new works cited:

[45] Kemna, E. W., Schoeman, R. M., Wolbers, F., Vermes, I., Weitz, D. A., & Van Den Berg, A. (2012). High-yield cell ordering and deterministic cell-in-droplet encapsulation using Dean flow in a curved microchannel. *Lab on a Chip*, 12(16), 2881-2887.

[46] Harrington, J., Esteban, L. B., Butement, J., Vallejo, A. F., Lane, S. I., Sheth, B., ... & West, J. (2021). Dual dean entrainment with volume ratio modulation for efficient droplet co-encapsulation: extreme single-cell indexing. *Lab on a Chip*, 21(17), 3378-3386.

[47] Edd, J. F., Di Carlo, D., Humphry, K. J., Köster, S., Irimia, D., Weitz, D. A., & Toner, M. (2008). Controlled encapsulation of single-cells into monodisperse picolitre drops. *Lab on a Chip*, 8(8), 1262-1264.

Reviewer comment 5: Point 2. *The technology is now described as allowing sparing use of precious samples. Please can the authors expand on sample processing. How do you preserve the remainder of the tissue/what losses occur in tubing, syringes, excess droplets. DropBlot only needs 5000 droplets, or approx. 1 microlitre at 4×10^6 cells, yet handling such small samples for microfluidic applications has not, as far as I am aware, been reported. The manuscript later reports that 100's of chemically fixed cells to be analysed.*

Author’s response 5: To address this helpful clarification request, we revise the manuscript to clarify that design is for the analysis of sparing specimens versus design for slab-gel assays. Further, we clarify the important difference between design for analysis of *sparing cell samples* versus design for *rare cells* (e.g., circulating tumor cells, CTCs):

The **Introduction** is revised to read: “As a corollary design goal, the DropBlot assay is designed to limit consumption of sparingly available archived, fixed biospecimens, hence the use of microfluidic design versus slab-gel assay formats. While not the subject of this study, the DropBlot reported here aims to create a foundation for extension of the DropBlot to analysis of rare cells, such as our previously reported single-cell immunoblotting tools for analysis of patient-derived circulating tumor cells (CTCs)³⁷ and individual cells (blastomeres) derived from single two-cell and four-cell murine embryos³⁸⁻⁴⁰.”

The **concluding sentence** of the manuscript is revised to read: “In terms of system design, while DropBlot is utilized here for serial, complementary unit functions (sample preparation to analysis or analysis to

detection), we see the modular design as potentially quite powerful for multi-omics questions where coupling of different, complementary analysis stages performed on the same individual cell may lead to breakthroughs in understanding. We also see an opportunity to redesign DropBlot for the handling and analysis of rare cells.”

Addition of new works cited:

[37] Sinkala, E., Sollier-Christen, E., Renier, C., Rosas-Canyelles, E., Che, J., Heirich, K., ... & Herr, A. E. (2017). Profiling protein expression in circulating tumour cells using microfluidic western blotting. *Nature communications*, 8(1), 14622.

[38] E Rosàs-Canyelles*, AJ Modzelewski*, A Geldert, L He, AE Herr, “Assessing heterogeneity among single embryos and single blastomeres using open microfluidic design”, *Science Advances*, 2020, 6(17):eaay1751. doi: 10.1126/sciadv.aay1751.

[39] E. Rosàs-Canyelles, A. J. Modzelewski, A. Geldert, L. He, A. E. Herr, “Multimodal detection of protein isoforms and nucleic acids from low starting cell numbers”, 2021, *Lab Chip*, 21(12):2427-2436. doi: 10.1039/d1lc00073j.

[40] E. Rosàs-Canyelles, A. J. Modzelewski, A. Geldert, L. He, A. E. Herr, “Multimodal detection of protein isoforms and nucleic acids from mouse preimplantation embryos”, *Nature Protocols*, 2021, 16(2):1062-1088. doi: 10.1038/s41596-020-00449-2.

Reviewer comment 6: Point 3. *Thank you for clarifying that urea methods are the topic of future assay developments.*

Author’s response 6: Thank you.

Response to Reviewer 3

Reviewer’s preface comment: *The authors have done significant work to revise the manuscript to address queries, and the addition of cell gating in clinical specimens is a valuable demonstration of the potential of this technology.*

Authors’ preface comment response: We thank the Reviewer for their enthusiasm regarding the cell-gating method.

Reviewer comment 1: *Some of the newly added text is excessively verbose, very detailed and could be edited for brevity and clarity*

Author’s response 1: We strive to balance succinctness against all three Reviewers’ requests for additional detail. In this second revision, we hope the Reviewer is satisfied with our redoubled efforts towards this goal.

Reviewer comment 2: *I have several comments regarding Figure 7:*

- I am concerned that some conclusions are based on very small numbers. For instance, the rare subpopulation of cells (“DropBlot detected a rare cell sub-population of ~0.4%...”) equates to 3 cells, if I understand correctly. This is simply too few to make a conclusion on, and I recommend removing reference to this in the results and abstract.

Author’s response 2: We appreciate the Reviewer’s suggestion and agree that the observation does not rise to the level of importance merited by the proposed inclusion in the **Abstract**. We revise the **Abstract** and **Result and Discussion** sections. Further, we include the single-cell data from the rare-cell subpopulation as **Figure S17** and reference from the main text.

Page 2 of **Abstract** removes mention of the uncommon profiles and now reads as: "...DropBlot of PFA-fixed human-derived breast tumor specimens (n = 5) showed antigen retrieval from cells archived frozen for 6 yrs. In this pilot study, the proteoform-resolving capacity of DropBlot revealed a rare subpopulation of VIM proteoform co-expression in HER2+ cells derived from an invasive ductal carcinoma tissue specimen. DropBlot could provide a precision integrated workflow for single-cell resolution protein-biomarker mining of precious biospecimen repositories."

Page 22 of **Result and Discussion** now reads: "Returning to Sample #3, the 707 cells analyzed were composed of 8.5% HER2+ cells (n = 60 cells, total: 707) by DropBlot. DropBlot detected HER2+ with no co-expression of VIM in ~5.4% (n = 38 cells, total: 707) of the cells assayed. Nearly ~2.7% (n = 19 cells, total: 707) of HER2+ cells expressed one but not the other VIM proteoform (1.3% VIM' only (n = 9 cells) and 1.4% VIM'' only (n = 3 cells, total: 707)). In addition, DropBlot detected three instances among the analyzed cell population (or 0.4% (n = 3 cells, total: 707)) that expressed HER2+ and also co-expressed both the VIM' and VIM'' proteoforms (Figure 7e, Figure S17)."

The **Supplemental Information** includes a new **Figure S17**:

Figure S17: Micrographs report that DropBlot detects co-expression of both VIM proteoforms (VIM', VIM'') in a small subpopulation of the HER2+ detection events from Patient Sample #3.

Reviewer comment 3: Please indicate the cell numbers that support each conclusion discussed.

Author's response 3: A fantastic suggestion! We fully agree and revise accordingly at several positions in the **Figure 7** labeling and surrounding discussion. Further, we add a summary **Table S6** of cell numbers to the SI for convenience:

Page 21 of **Result and Discussion** now reads: "...Attending to Sample #3 in a moment, Samples #1 and #2 were fresh cell suspensions and contained 26.4% (n = 527 cells among 1996 cells analyzed) and 38.9% (n = 474 cells among 1219 cells analyzed) HER2+ cells, respectively. For convenience, **Table S6** reports the sample size of each starting cell population and each respective subpopulation. While 0% (n = 0 cells, total: 1996) of Sample #1 cells and 7.5% (n = 91 cells, total: 1219) of Sample #2 cells co-expressed VIM', there were no cells detectable that expressed the VIM'' proteoform (either alone or with VIM). Samples #4 and #5, which were suspensions of fresh dissociated tissues, showed 46.3% (n = 925 cells, total: 1998) and 0% cells as HER2+ (n = 0 cells, total: 1442), respectively. In Sample #4, 5.4% (n = 108 cells, total: 1998) HER2+ cells co-expressed VIM+, with VIM expression arising only from the VIM' proteoform. VIM'' was not detected in Sample #4, either alone or co-expressed with VIM'. Sample #5 reported a cell sub-population of 41.9% (n = 604 cells, total: 1442) of the cells analyzed, which was EpCAM-/HER2- and was expressing VIM' alone and likely stromal in origin."

Page 22 of **Result and Discussion** now reads: “Returning to Sample #3, the 707 cells analyzed were composed of 8.5% HER2+ cells (n = 60 cells, total: 707) by DropBlot. DropBlot detected HER2+ with no co-expression of VIM in ~5.4% (n = 38 cells, total: 707) of the cells assayed. Nearly ~2.7% (n = 19 cells, total: 707) of HER2+ cells expressed one but not the other VIM proteoform (1.3% VIM’ only (n = 9 cells, total: 707) and 1.4% VIM’’ only (n = 10 cells, total: 707)). In addition, DropBlot detected three instances among the analyzed cell population (or 0.4% (n = 3 cells, total: 707)) that expressed HER2+ and also co-expressed both the VIM’ and VIM’’ proteoforms (Figure 7e, Figure S17).”

Page 22 of **Result and Discussion** now reads: “...In the VIM+ stromal cells of Sample #3, 30.8% (n = 218 cells, total: 707) expressed VIM’, 8.5% (n = 60 cells, total: 707) expressed VIM’’, and 10.3% (n = 73 cells, total: 707) co-expressed VIM’ and VIM’’ (Figure S18).”

In a novel function of the DropBlot assay which is gating on differential proteoform expression, we next gated the cell population on expression of just one of the VIM proteoforms (Figure S19, Figure S20). Just ~1.3% (n = 9 cells, total: 707) of the cells analyzed from Sample #3 co-expressed VIM’ with HER2 and ~9.6% (n = 68 cells, total: 707) co-expressed VIM’ with EpCAM. Next gating on expression of the second VIM’’ proteoform only, DropBlot identified ~1.4% (n = 10 cells, total: 707) of the cells analyzed from Sample #3 as co-expressing VIM’’ with HER2 and ~5.2% (n = 37 cells, total: 707) co-expressing VIM’’ with EpCAM. Roughly 17.1% (n = 121 cells, total: 707) of the Sample #3 cells analyzed were positive for VIM (VIM’ or VIM’’) and EpCAM, with ~2.3% (n = 16 cells, total: 707) expressing both VIM proteoforms and EpCAM.”

We revised the SI to include a summary **Table S6** collating the detection events reported in **Figure 7** and the surrounding discussion:

Table S6. Summary of cell subpopulations identified by DropBlot
(values represent the number of cells analyzed)

Subpopulation	VIM proteoforms	Sample #1	Sample #2	Sample #3	Sample #4	Sample #5
EpCAM+	VIM’-/VIM’’-	452	501	175	0	440
	VIM’+/VIM’’-	0	176	68	0	398
	VIM’-/VIM’’+	0	0	37	0	0
	VIM’+/VIM’’+	0	0	16	0	0
	Total_EpCAM	452	677	296	0	838
HER2+	VIM’-/VIM’’-	527	383	38	817	0
	VIM’+/VIM’’-	0	91	9	108	0
	VIM’-/VIM’’+	0	0	10	0	0
	VIM’+/VIM’’+	0	0	3	0	0
	Total_HER2	527	474	60	925	0
Other (EpCAM-/HER2-)	VIM’+/VIM’’-	845	68	218	1073	604
	VIM’-/VIM’’+	110	0	60	0	0
	VIM’+/VIM’’+	62	0	73	0	0
	Total_other	1017	68	351	1073	604
Number of cells analyzed		1996	1219	707	1998	1442

Figure 1. DropBlot analysis of single PFA- and methanol-fixed patient-derived dissociated cancer cells. (a) Schematic of clinical sample preparation workflow for DropBlot. PFA conditions: $\Delta t_{\text{fixation}} = 15 \text{ min}$; $\Delta t_{\text{incubation}} = 1.0 \text{ h}$ at 98°C ; Figures generated with BioRender. (b) Fluorescence micrographs of single-cell western blots of EpCAM (Green, AF488-labeled secondary antibody), VIM (Red, AF594-labeled secondary antibody), and HER2 (Blue, AF647-labeled secondary antibody) from PFA-fixed tumor cell. Tumor was classified as triple-positive breast cancer. (c) Mean fluorescence intensity of single-cell western blot analyses of PFA-fixed, patient-derived tumor cells for EpCAM, VIM proteoforms (VIM', VIM''), and HER2. Samples #1-2 were fresh cell suspensions. Sample #3-5 were fresh dissociated tissues. The tumor cells were identified as EpCAM+ or HER2+. (d) Fluorescence micrographs of single-cell western blots of PFA-fixed cells from Sample #3 from (c), with HER2 (blue, AF647-labeled secondary antibody) and VIM (red, AF594-labeled secondary antibody). (e) Cell gating using DropBlot. HER2+ positive cells were further classified based on the expression levels of VIM' and VIM''. The protein target was considered as negative when the intensity was less than 10. Gate threshold intensity: 100; Background threshold intensity: 10. (f) Cell gating using DropBlot. EpCAM+ positive cells were further classified based on the expression levels of VIM' and VIM''. The protein target was considered as negative when the intensity was less than 10. Gate threshold intensity: 100; Background threshold intensity: 10. (g) Venn diagram reporting the single-cell target-expression profile for each of single PFA-fixed cells from Sample #1-5 in (c). See Table S6 for cell enumeration of each subpopulation.

Reviewer comment 4: Please show histograms for Her2 and EPCAM alone, with a line showing the threshold used for gating. Why do the plots in 7e & 7f look as though they have been thresholded? Ie, there is not a

continuous distribution on either axis. Please show unfiltered histograms for VIM isoforms, both with and without pre-gating for ERBB2/EPCAM

Author's response 4: Correct, micrograph background subtraction is necessary, as described in the Methods. To make the protocol more obvious to a reader, we revised **Figure 7** and added **Figure S19** and **Figure S20** to report the histograms both *with* and *without* gating for HER2, EpCAM, and VIM isoforms per the Reviewer's suggestion.

Figure 2. DropBlot analysis of single PFA- and methanol-fixed patient-derived dissociated cancer cells. (a) Schematic of clinical sample preparation workflow for DropBlot. PFA conditions: $\Delta t_{\text{fixation}} = 15 \text{ min}$; $\Delta t_{\text{incubation}} = 1.0 \text{ h}$ at 98°C ; Figures generated with BioRender. (b) Fluorescence micrographs of single-cell western blots of EpCAM (Green, AF488-labeled secondary antibody), VIM (Red, AF594-labeled secondary antibody), and HER2 (Blue, AF647-labeled secondary antibody) from PFA-fixed tumor cell. Tumor was classified as triple-positive breast cancer. (c) Mean fluorescence intensity of single-cell western blot analyses of PFA-fixed, patient-derived tumor cells for EpCAM, VIM proteoforms (VIM', VIM''), and HER2. Samples #1-2 were fresh cell suspensions. Sample #3-5 were fresh dissociated tissues. The tumor cells were identified as EpCAM+ or HER2+. (d) Fluorescence micrographs of single-cell western blots of PFA-fixed cells from Sample #3 from (c), with HER2 (blue, AF647-labeled secondary antibody) and VIM (red, AF594-labeled secondary antibody). (e) Cell gating using DropBlot. HER2+ positive cells were further classified based on the expression levels of VIM' and VIM''. The protein target was considered as negative when the intensity was less than 10. Gate threshold intensity: 100; Background threshold intensity: 10. (f) Cell gating using DropBlot. EpCAM+ positive cells were further classified based on the expression levels of VIM' and VIM''. The protein target was considered as negative when the intensity was less than 10. Gate threshold intensity: 100; Background threshold intensity: 10. (g) Venn diagram reporting

the single-cell target-expression profile for each of single PFA-fixed cells from Sample #1-5 in (c). See Table S6 for cell enumeration of each subpopulation.

The Supplemental Information includes a new Figure S19 and S20:

Figure S19. Histogram of EpCAM and HER2 intensity in patient samples. Gate threshold intensity: 100; background threshold intensity: 10.

Figure S20. Histograms of VIM' & VIM'' intensity before and after HER2/EpCAM gating. The light green boxes in the middle row indicate EpCAM+ cells after gating, whereas the blue boxes in the bottom row indicate HER2+ cells after gating. Gate threshold intensity: 100; background threshold intensity: 10.

Page 22 of **Result and Discussion** now reads: “In a novel function of the DropBlot assay which is gating on differential proteoform expression, we next gated the cell population on expression of just one of the VIM proteoforms (**Figure S19, Figure S20**).”

Reviewer comment 5: *Regarding this statement “Sample #5 reported a cell sub-population of 41.9% of the cells analyzed, which was expressing VIM’ alone and likely stromal in origin”: Please explain this further. Does this mean 41.9% of cells were Her2+Vim+? If so, stromal cells are typically not Her2+. Please explain*

Author’s response 5: Thank you for pointing out the confusing wording. These VIM’ expressing cells were not expressing detectable levels of HER2 nor EpCAM, we revise the **Result and Discussion** to:

Page 21 of **Result and Discussion** now reads: “Sample #5 reported a cell sub-population of 41.9% (n = 604 cells, total: 1442) of the cells analyzed, which was EpCAM-/HER2- and was expressing VIM’ alone and likely stromal in origin.”

Reviewer comment 6: *Regarding this statment: “Returning to Sample #3, the 707 cells analyzed were composed of 8.5% HER2+ cells by DropBlot. DropBlot detected HER2+ with no co-expression of VIM in ~5.4% of the cells assayed”: In the figure it appears to be 63.3% rather than 5.4%. Please clarify*

Author’s response 6: Thank you. To avoid confusion regarding starting sample size, we revise **Figure 7e-f** labeling and add **Table S6** (mentioned above and in Figure 7 caption) to clarify that the percentage reported was calculated based on the total number of cells analyzed:

Figure 3. DropBlot analysis of single PFA- and methanol-fixed patient-derived dissociated cancer cells. (a) Schematic of clinical sample preparation workflow for DropBlot. PFA conditions: $\Delta t_{\text{fixation}} = 15 \text{ min}$; $\Delta t_{\text{incubation}} = 1.0 \text{ h}$ at 98°C ; Figures generated with BioRender. (b) Fluorescence micrographs of single-cell western blots of EpCAM (Green, AF488-labeled secondary antibody), VIM (Red, AF594-labeled secondary antibody), and HER2 (Blue, AF647-labeled secondary antibody) from PFA-fixed tumor cell. Tumor was classified as triple-positive breast cancer. (c) Mean fluorescence intensity of single-cell western blot analyses of PFA-fixed, patient-derived tumor cells for EpCAM, VIM proteoforms (VIM', VIM''), and HER2. Samples #1-2 were fresh cell suspensions. Sample #3-5 were fresh dissociated tissues. The tumor cells were identified as EpCAM+ or HER2+. (d) Fluorescence micrographs of single-cell western blots of PFA-fixed cells from Sample #3 from (c), with HER2 (blue, AF647-labeled secondary antibody) and VIM (red, AF594-labeled secondary antibody). (e) Cell gating using DropBlot. HER2+ positive cells were further classified based on the expression levels of VIM' and VIM''. The protein target was considered as negative when the intensity was less than 10. Gate threshold intensity: 100; Background threshold intensity: 10. (f) Cell gating using DropBlot. EpCAM+ positive cells were further classified based on the expression levels of VIM' and VIM''. The protein target was considered as negative when the intensity was less than 10. Gate threshold intensity: 100; Background threshold intensity: 10. (g) Venn diagram reporting the single-cell target-expression profile for each of single PFA-fixed cells from Sample #1-5 in (c). See Table S6 for cell enumeration of each subpopulation.

Reviewer comment 7:- The paragraph commencing “In a novel function of the DropBlot assay, we next gated the cell population on expression of just one of the VIM proteoforms.” is confusing and does not appear to have a focus. I suggest either deleting it or clarifying, including cell numbers in each population discussed

Author's response 7: We appreciate the suggestion, and opt to revise the referenced paragraph because the analysis highlights DropBlot's ability to gate on differential proteoform detection. Following the Reviewer's suggestion, we revise the rationale in the **thesis sentence**, and now include cell numbers in each population discussed and have added **Table S6** to enumerate cell numbers analyzed.

Page 22 of **Result and Discussion** now reads: "In a novel function of the DropBlot assay **which is gating on differential proteoform expression**, we next gated the cell population on expression of just one of the VIM proteoforms (**Figure S19, Figure S20**). Just ~1.3% (**n = 9 cells, total: 707**) of the cells analyzed from Sample #3 co-expressed VIM' with HER2 and ~9.6% (**n = 68 cells, total: 707**) co-expressed VIM' with EpCAM. Next gating on expression of the second VIM'' proteoform only, DropBlot identified ~1.4% (**n = 10 cells, total: 707**) of the cells analyzed from Sample #3 as co-expressing VIM'' with HER2 and ~5.2% (**n = 37 cells, total: 707**) co-expressing VIM'' with EpCAM. Roughly **17.1% (n = 121 cells, total: 707)** of the Sample #3 cells analyzed were positive for VIM (VIM' or VIM'') and EpCAM, with ~2.3% (**n = 16 cells, total: 707**) expressing both VIM proteoforms and EpCAM."

Reviewer comment 8: *A general comment is that the single cell results would be more convincing if bulk immunoblotting data was provided to refer to for each sample analysed*

Author's response 8: We agree. Unfortunately, the limited amount of material extractable from the patient-derived solid primary tumor specimens analyzed by DropBlot precluded bulk analyses and DropBlot assay development on the same specimens. As we mature this first-report of the DropBlot assay towards specific clinical and biological questions, as well as potentially clinical trial inclusion, side-by-side conventional and single-cell assays will be performed when dual analyses of human specimens prove feasible.

REVIEWERS' COMMENTS

Reviewer #2 (Remarks to the Author):

The authors have addressed my comments on (1) the channel geometry for cell ordering; (2) how the loading statistics were produced; (3) the actual cell numbers analysed; (4) reference background for single cell encapsulation methods.

Comment 5 deserves explanation and clarification. While the purpose of the method is now clarified (for use with sparing cell numbers), practical details are required to explain how it is possible to manage small cell numbers in high concentrations. How are such small volumes prepared and delivered to the device? Or is it the case that many more cells are prepared from the fixed section, and those left over can be retrieved for archive/further analysis.

Reviewer #3 (Remarks to the Author):

The authors have addressed my queries and I am satisfied with this manuscript. Congratulations on this work.

Response to Reviewer #2

Reviewer comment 1: *The authors have addressed my comments on (1) the channel geometry for cell ordering; (2) how the loading statistics were produced; (3) the actual cell numbers analysed; (4) reference background for single cell encapsulation methods.*

Author's response 1: We thank the reviewer for their great suggestions.

Reviewer comment 2: *Comment 5 deserves explanation and clarification. While the purpose of the method is now clarified (for use with sparing cell numbers), practical details are required to explain how it is possible to manage small cell numbers in high concentrations. How are such small volumes prepared and delivered to the device? Or is it the case that many more cells are prepared from the fixed section, and those left over can be retrieved for archive/further analysis.*

Author's response 2: We thank the reviewer for requesting clarification. We revise the description to list the potential methods to manage small-cell numbers.

Page 24 of the **Results and Discussion** is revised to read: "...While promising for the dual design goals of (i) analysis of sparingly limited cell specimens from biorepositories and (ii) proteoform detection, several areas for obvious performance enhancement and optimization exist, depending on the application area of interest. Future work will include the integration of established sample preparation techniques, such as empty droplet depletion⁸¹ and single-particle trapping techniques⁸², to detect rare cells. With a keen focus on preparation and analyses of single cells derived from dissociated solid tumor specimens, cells that are not analyzed can be retrieved for archiving and future analysis. A further area entails expanding target-detection multiplexing beyond the handful of targets detected in each cell that is reported in this **first** report..."

Addition of new works cited:

[81] De Jonghe, J., Kaminski, T. S., Morse, D. B., Tabaka, M., Ellermann, A. L., Kohler, T. N., ... & Hollfelder, F. (2023). spinDrop: a droplet microfluidic platform to maximise single-cell sequencing information content. *Nature Communications*, 14(1), 4788.

[82] Li, X., & Lee, A. P. (2018). High-throughput microfluidic single-cell trapping arrays for biomolecular and imaging analysis. In *Methods in cell biology* (Vol. 148, pp. 35-50). Academic Press.

Response to Reviewer #3 (Remarks to the Author):

Reviewer comment: *The authors have addressed my queries, and I am satisfied with this manuscript. Congratulations on this work.*

Author's Response: Thank you – we are excited about this new technique, as well. Further, we believe that the manuscript is stronger in revision owing to your questions and comments.